

# Golden eagle optimized CONV-LSTM and non-negativity-constrained autoencoder to support spatial and temporal features in cancer drug response prediction

Wesam Ibrahim Hajim[1,2], Suhaila Zainudin[2], Kauthar Mohd Daud[2] and Khattab Alheeti[3]

[1] Department of Applied Geology, College of Sciences, University of Tikrit, Tikrit, Salah ad Din, Iraq
[2] Center for Artificial Intelligence Technology, Faculty of Information Science and Technology, Universiti Kebangsaan Malaysia, Selangor, Malaysia
[3] Department of Computer Networking Systems College of Computer Sciences and Information Technology, University of Anbar, Ramadi, Al Anbar, Iraq

Corresponding author
Wesam Ibrahim Hajim,
p106883@siswa.ukm.edu.my

## ABSTRACT

Advanced machine learning (ML) and deep learning (DL) methods have recently been utilized in Drug Response Prediction (DRP), and these models use the details from genomic profiles, such as extensive drug screening data and cell line data, to predict the response of drugs. Comparatively, the DL-based prediction approaches provided better learning of such features. However, prior knowledge, like pathway data, is sometimes discarded as irrelevant since the drug response datasets are multidimensional and noisy. Optimized feature learning and extraction processes are suggested to handle this problem. First, the noise and class imbalance problems must be tackled to avoid low identification accuracy, long prediction times, and poor applicability. This article aims to apply the Non-Negativity-Constrained Auto Encoder (NNCAE) network to tackle these issues, enhance the adaptive search for the optimal size of sliding windows, and ensure that deep network architectures are adept at learning the vital hidden features. NNCAE methodology is used after performing the standard pre-processing procedures to handle the noise and class imbalance problem. This class balanced and noise-removed input data features are learned to train the proposed hybrid classifier. The classification model, Golden Eagle Optimization-based Convolutional Long Short-Term Memory neural networks (GEO-Conv-LSTM), is assembled by integrating Convolutional Neural Network CNN and LSTM models, with parameter tuning performed by the GEO algorithm. Evaluations are conducted on two large datasets from the Genomics of Drug Sensitivity in Cancer (GDSC) repository, and the proposed NNCAE-GEO-Conv-LSTM-based approach has achieved 96.99% and 97.79% accuracies, respectively, with reduced processing time and error rate for the DRP problem.

## INTRODUCTION

Cancer is a disease that occurs due to the abnormal proliferation of cells and can spread to other body parts. According to a World Health Organization (WHO) report, cancer accounted for about 10 million deaths in 2020, or nearly one in six deaths, making it the leading cause of death worldwide. Breast, lung, colon, rectum, and prostate cancers are the most commonly encountered types of cancer (*Miller et al., 2022*). The two most prevalent invasive cancers, respectively, are prostate cancer, which is prostate adenocarcinoma (GDSC2) in men (*Li et al., 2020*), and breast cancer (GDSC1) in women (*Pucci, Martinelli & Ciofani, 2019*). Many treatment methods have been developed to treat cancer, resulting in an increased number of databases (*Debela et al., 2021*). Therefore, special attention should be paid to selecting a reliable database to evaluate the drug response prediction (DRP) model, such as the GDSC, Cancer Cell Line Encyclopedia (CCLE), and The Cancer Genome Atlas (TCGA) (*Naqvi, Rizvi & Hassan, 2023*).

The application of predictive modeling exhibits significant potential in enhancing individualized cancer treatment and optimizing the efficacy of drug development (*Tufail et al., 2024*). Recently, the number of studies utilizing the DL algorithms for the DRP process has been increasing tremendously, focusing on preprocessing and feature learning, which has demonstrated superior performance compared to traditional machine learning in terms of generalizing predictions to new data. However, the techniques have limitations in incorporating the cell line, feature learning, dimensionality reduction, data classification, and multidimensional features, leading to low prediction performance (*Chugh, Kumar & Singh, 2021*).

Cancer data present intrinsic noise, and DL-based DRP models often suffer from noise present in the data. Noisy data can reduce the accuracy of a DL model, as the model may be unable to distinguish between the signal and the noise, leading to an unnecessary increase in the complexity of the model (*Pepe et al., 2022*). Noise can result from laboratory equipment, problems in the storage of the material, and contaminations of specimen manipulation by researchers (*Baptista, Ferreira & Rocha, 2021*; *Xia et al., 2022*).

DL-based prediction models can handle noisy datasets, but inappropriate feature extraction methods might induce class imbalance problems (*Kafunah, Ali & Breslin, 2021*). In addition, the noisy dataset and the sub-optimal values for the hyperparameters of the classifier will make it inadequate for learning multidimensional datasets and diminish the overall prediction accuracy (*Castillo et al., 2020*). This will cause problems such as data imbalance, reduced prediction accuracy, and increased training periods.

Considering these challenges, the main objective of this research is to develop a new DL model that integrates recently developed meta-heuristics with deep convolutional neural networks to optimize their architectures. This will enhance existing DL methods by providing the algorithm with automatic methods to deal with intrinsic noise.

This article describes an efficient DRP framework using two advanced deep-learning techniques. The initial technique, known as the NNCAE, improves the accuracy of cancer drug response prediction by detecting positive associations between the molecular properties of cancer cells and the effectiveness of drugs. Negative features refer to the

aspects of the data or characteristics that could disrupt the learning process or negatively impact the prediction performance of the DRP model. These negative features include noise, class imbalance, and irrelevant or redundant information in genomic and drug response data.

*Foersch et al. (2023)* employed a DL model, specifically a multi-stain deep learning model (MSDLM), and analyzed the immune cell composition in Colorectal cancer (CRC) tumors. The model utilized individuals with rectal cancer and neoadjuvant therapy and forecasted the results. However, the model lacks interpretability and training time and depends on cohorts or distinct cancer types. *Foersch et al. (2023)* developed a novel DL-based method called DeepInsight-3D for predicting patient-specific anti-cancer drug responses from multi-omics data. For feature selection and classification, the model used a ResNet-50 CNN architecture that has been trained. Hyperparameters such as momentum, L2 regularization, and initial learning rate are used for DeepInsight. The hyperparameters are adjusted using a Bayesian optimization technique and enhanced prediction performance. Across seven drug response datasets, DeepInsight-3D produced an average area under the curve Area Under Curve (AUC) of 0.72. However, the model has limited availability of training and test samples, which can affect model estimation and confidence. *Chen et al. (2023)* presented a parallel DL network framework called DNN-PNN to predict anti-cancer drug sensitivity by integrating gene expression data and pharmaceutical chemical structure data. The model combined heterogeneous data from several sources for improved prediction and addressed dimensionality and sparsity problems. However, the model has no quantitative results and metrics reported to gauge performance improvements.

*Liu et al. (2024)* proposed a Multi-task Interaction Graph Convolutional Network (MTIGCN) for predicting anti-cancer drug response. MTIGCN combined drug sensitivity classification, IC50 regression prediction, and similarity network reconstruction tasks to enhance feature representations and reduce overfitting. However, the computational complexity limits the model's scalability to larger datasets or more complex tasks. *Zhou et al. (2024)* proposed a Multi-omics Fusion Graph Attention Network (MFGAN) model for survival prediction and DRP in digestive system tumors using multi-omics data. The proposed MFGAN model demonstrated improved performance, up to a 9% improvement in the c-index metric for survival prediction and a 4% improvement in DRP performance. However, the model has a limited scope for digestive system tumors. *Zheng et al. (2024)* proposed a multi-omics data classification network called the Global and Cross-Modal Feature Aggregation Network (GCFANet) that integrates complementary information from different modalities while capturing sample structure and feature confidence. However, the model lacks the interpretability of the learned feature representations and consensus representations.

The standard AE has redundant representations, which use more elements than necessary to describe the data, increasing model complexity and computation time. To reduce the complexity and processing time, constraining properties can be controlled (*Zhang, Chen & Li, 2021*). The proposed NNCAE introduces nonnegativity as a constraint,

which prevents biased learning and improves the extraction of meaningful representations for learning the hidden structure of high-dimensional data.

The second method, (GEO-Conv-LSTM), is developed by integrating CNN and LSTM networks to form a hybrid classifier with its parameters optimized using GEO. This model was created using the combination of CNN and LSTM framework (*Qian et al., 2023*) to learn spatial and temporal features of gene expression data. The gene expression data collected based on the size of the cancer regions after the medication are termed spatial features. The features collected at particular periods are termed temporal features in gene expression data. The CNN model increases the prediction accuracy, and the LSTM model minimizes the training time. The hyperparameter of this model is tuned by the GEO technique to reduce the convergence problem and model complexity of this hybrid model. The generated feature subsets by NNCAE are used to train the hybrid model to predict drug response in breast and prostate cancer datasets. The contribution of this work is described as follows:

- NNCAE is presented to remove the noisy, outlier data features and balance the minority and majority class samples. It is developed by integrating the nonnegativity constraints into the denoising autoencoder with an effective learning structure for the temporal features. Implementing the NNCAE model also solves the class imbalance problem.
- GEO-Conv-LSTM is used to learn the features from gene expression, mutation profiles, and drug data to improve the prediction accuracy of anti-cancer drug responses. It is developed by integrating GEO for optimally tuning the hybrid Conv-LSTM classifier framework's hyper-parameters, whose architecture combines the CNN and LSTM layers. The temporal and spatial features are learned by the Conv-LSTM framework, which enhances the prediction performance and decreases the training time.
- Experiments are conducted over the GDSC1 and GDSC2 multi-omics datasets from GDSC.

## RELATED WORKS

This review of the large body of literature revealed many studies that used different techniques to predict drug responses. Below is a summary of a selection of these studies.

*Mohamed Salleh et al. (2015)* proposed a method to improve Gene Regulatory Network (GRN) inference accuracy from gene expression data by combining the Gaussian noise model and the Pearson correlation coefficient. *Hosseini-Asl, Zurada & Nasraoui (2016)* introduced a DL auto-coder network trained using an (NNCAE), resulting in improved sparsity and reconstruction quality compared to traditional autoencoders.

*Mohamed, Zainudin & Ali Othman (2017)* proposed a metaheuristic strategy to improve the mRMR filter method in drug response microarray data categorization using three metaheuristic algorithms: Particle Swarm Optimization (PSO), cuckoo search (CS), and artificial bee colony (ABC) *Husam et al. (2017)* introduced a new way for Malaysian health agencies to find the best features for more accurate predictions of dengue outbreaks. They used three feature selection algorithms: PSO, genetic algorithm (GA), and rank

search (RS). The experimental results showed that the feature selection process improved predictive accuracy, achieving an accuracy of 98.85.

*Salleh, Zainudin & Arif (2017)* proposed using Multiple Linear Regression (MLR) to infer the GRN from gene expression data and avoid wrongly inferring indirect interactions. Experiments show that MLR makes significantly fewer cascade errors (<10%) when predicting the subnetworks with added cascade motifs. All the tested subnetworks obtained satisfactory results, with AUROC values above 0.5. *Ananda, Daud & Zainudin (2023)* provided a comprehensive review of regulatory-metabolic network models, which integrate GRNs and metabolic networks to optimize microbial strains to produce valuable compounds. These models' performance heavily depends on the quality and quantity of the gene expression data used. Inconsistencies between the GRNs and the gene expression data can lead to inaccurate predictions.

*Chiu et al. (2019)* implemented a deep neural network (DNN) methodology for predicting drug responses, with a median MSE of 1.96 obtained in testing instances. *Sharifi-Noghabi et al. (2019)* developed a multi-omics late integration model based on DNN for DRP with a prediction accuracy of 0.80 for the drug cetuximab. *Shi et al. (2019)* predicted drug-target interactions using Lasso and Random Forest techniques, with an overall accuracy of 0.99% for the Drug Bank dataset. *Kajikawa et al. (2019)* presented a dose distribution prediction methodology using a three-dimensional CNN model with more effective results than Rapid Plan but increased computational complexity. *Xu et al. (2019)* used Autoencoder (AE), the Boruta algorithm, and Random Forest (RF) techniques to predict drug response.

*Yu et al. (2021)* applied discrete component analysis (DCA) and logistic regression (LR) methodologies to predict drug responses and achieved an overall accuracy of 0.9107. *Lee & Chen (2021)* developed a hybrid model that integrates graph convolutional neural networks (GCNN) and Bi-directional LSTM networks to predict drug side effects. *Malik, Kalakoti & Sundar (2021)* presented a multi-omics integrative framework based on a feed-forward neural network for quantifying the survival and drug response of GDSC1 patients. They used Neighbourhood Component Analysis (NCA) to eliminate irrelevant features and achieved an accuracy of 94% for predicting cancer subtypes. *Zhang et al. (2021)* developed a drug combination prediction model using a DNN with AEs. They achieved an accuracy of $0.93 \pm 0.01$ compared to GBM, Elastic Nets, Deepsynergy, and RF models.

*Zhang, Chen & Li (2021)* designed a deep signaling framework for predicting the anti-drug response. They used 46 signaling pathways for DRP models and achieved higher performance than DNN and other conventional methods. *Majumdar et al. (2021)* suggested K-means Ensemble Support Vector Regression (KESVR) for DRP. However, this model predicted drug responses from a single perspective, leading to inappropriate predictions. *Singh et al. (2021)* propose a beetle swarm optimization and adaptive neuro-fuzzy inference system (BSO-ANFIS) model. The model uses a modified Crow search algorithm for feature extraction and an ANFIS classification model optimized by a BSO algorithm. The algorithm achieves 99:1% accuracy in heart disease detection and 96:08% accuracy in multi-disease classification.

*Kamal, Bakar & Zainudin (2022)* presented a study addressing ineffective protein feature representation in hierarchical classification structures. This method combines PSO and the firefly algorithm (FAPSO) by using discrete wavelet transform (DWT) to improve the representation of features. The efficacy of this methodology was validated using a G protein-coupled receptors (GPCR) dataset; the study achieved classification accuracies of 97.9%, 86.9%, and 81.3%, respectively, at the family, subfamily, and sub-subfamily levels.

*Jiang et al. (2022)* introduced the DeepTTA model, which analyzed drug chemical substructure and gene expression data to predict drug responses with higher accuracy. This model found dactinomycin and bortezomib as potential therapeutic options but had high computational complexity. *Wang et al. (2022)* presented the Multi-View Multi-Omics (MVMO) model, which resolved missing value problems in gene expression data and used fewer parameters and less DRP computation than existing approaches. *Zhu et al. (2022)* introduced the Twin Graph Neural Networks Similarity Augmentation (TGSA) model, which combined Twin Graph neural networks and similarity augmentation modules for drug response prediction. *Ogunleye et al. (2022)* recommended the Classification and Regression Tree (CART) model for DRP, which gained higher Matthews correlation coefficient (MCC) values and AUC values using the TCGA-GDSC1 dataset.

*Wu, Zainudin & Daud (2023)* have presented a new study discussing GCNs for drug repositioning. GCNs have shown promising results in extracting features from heterogeneous graphs for drug repositioning. By using more than one feature mining technique, such as similarity calculations, and other semi-supervised learning models, such as LSTM, GCNs can be better at extracting information for drug repositioning. They also aim to combine GCNs and CNNs to mine drug association data and extract more features, such as the CNN-based multi-scale interaction feature fusion method for drug prediction. Although the study was extensive, it needed help finding a satisfactory solution to address the overfitting problem using the similarity-based heterogeneous graph inference approach.

*Hostallero et al. (2023)* developed a deep learning framework called TINDL, thoroughly trained on preclinical cancer cell lines (CCLs), which can predict the response of cancer patients to different treatments and identify biomarkers of drug response. TINDL outperforms other methods in distinguishing between sensitive and resistant patients for 10 out of 14 drugs. The primary outcome measured in this study was the prediction of CDR in cancer patients. In addition, identification of biomarkers of drug response. The models were trained on single-drug responses, but the test data included patients who received multiple drugs, which the models may not accurately predict. The use of cell line data for training may limit the ability of the models to capture the complexity of actual tumor samples.

*Kato et al. (2023),* provided a comprehensive review of deep learning methods for predicting DRP. This article conducted an extensive search and analysis of 61 peer-reviewed publications on DL models for predicting drug response in cancer. The article identified three major components involved in developing DRP models: data preparation, model development, and performance analysis. Some limitations have been identified, such as the lack of a standard or accepted framework for evaluating and comparing cancer

drug response models. Most models focus on improving predictions in cell lines with only marginal improvements in generalization to other cancer models. In addition, there are concerns about the interpretability and transparency of black-box DL models for clinical decision-making.

*Teoh et al. (2024)* developed an ensemble model, combining ResNet-50 and GoogLeNet to accurately classify microcalcifications in breast cancer images. The model achieved an average confidence level of 0.9305 in microcalcification classification and 0.8859 in normal cases classification. Its performance was compared to other models, showing high accuracy, recall, and precision with an AUC of 99.06%. However, the model's current method of handling missing data and not fully capturing relevant features may limit its predictive capabilities.

*Daud et al. (2023)* provided a systematic review, this review article aims to provide current trends and approaches in machine learning and constraint-based modeling for *in silico* metabolic engineering and highlight research gaps in this area. The authors conducted a systematic review of relevant studies from the Web of Science and Scopus databases and synthesized and analyzed 13 relevant studies that integrated 17 different machine-learning approaches with constraint-based modeling. Even though the study covered many models, more in-depth research and biological validation studies are needed to make machine learning better at predicting phenotypic changes and making them easier to understand and use.

*Cao, Zainudin & Daud (2024)* was able to fuse features using FFANE, a novel node representation method that combines protein-protein interactions (PPIs) networks and protein sequence data using Gaussian kernel, Levinstein distance, and Stacked Autoencoder (SAE) to enhance PPI prediction. This proposed method achieved high prediction accuracies of 94.28%, 97.69%, and 84.05% on *Saccharomyces cerevisiae*, *Homo sapiens*, and *Helicobacter pylori* protein-protein interaction datasets, respectively. Despite the novelty of the method, FFANE has modest hardware requirements compared to deep learning models, which can be considered a limitation in terms of not taking advantage of the latest advances in computational power.

*Hajim et al. (2024)* provided a comprehensive review of deep learning models for drug response prediction in cancer therapeutics. The study examined the latest developments in deep learning from 2017 to 2023. The study focused on neural networks, such as DNNs, recurrent neural networks (RNNs), CNNs, and supervised and unsupervised learning. The study underscores the need to improve the generalizability and interpretability of the models and the importance of addressing heterogeneity in cancer data to enhance their predictive accuracy and clinical validity.

This study highlighted the advancements in methodologies that use ML and DL techniques. The works previously described indicate the relevance of the theme involving computational methods to support pathologists in the study of cancer disease, despite the advancements in methodologies that used these techniques to enhance prediction accuracy and computational efficiency, which contributed to developing and improving the accuracy of DRP. However, several gaps and challenges emerge from this literature review, indicating areas for future research. Here is a synthesis of the identified gaps.

**Table 1 List of symbols and abbreviations.**

| Sample | Abbreviations |
| --- | --- |
| AI | Artificial Intelligence |
| ANFIS | Adaptive Neural Fuzzy Inference System |
| ABC | Artificial Bee Colony |
| AE | Autoencoder |
| ANNs | Artificial Neural Networks |
| AUC | Area Under Curve |
| AUROC | Area Under the Receiver Operating Characteristic Curve |
| BRCA | Breast Cancer |
| BSO-ANFIS | Beetle Swarm Optimization and Adaptive Neuro-Fuzzy Inference System |
| CART | Classification And Regression Tree |
| CCLE | Cancer Cell Line Encyclopedia |
| CNN | Convolution Neural Network |
| CRC | Colorectal Cancer |
| CS | Cuckoo Search |
| DCA | Discrete Component Analysis |
| DL | Deep Learning |
| DLNN | Deep Learning Neural Network |
| DNN | Deep Neural Network |
| DRP | Drug Response Prediction |
| DWT | Discrete Wavelet Transform |
| FA | Firefly Algorithm |
| FAPSO | Firefly Algorithm Particle Swarm Optimization |
| FCN | Fully Connected Network |
| GA | Genetic Algorithm |
| GCFANet | Cross-Modal Feature Aggregation Network |
| GCNN | Graph Convolutional Neural Networks |
| GDSC | Genomics Of Drug Sensitivity in Cancer |
| GEO | Golden Eagle Optimization |
| GPCR | G Protein-Coupled Receptors |
| GRN | Gene Regulatory Network |
| GRNN | Generalized Regression Neural Network |
| KESVR | K-Means Ensemble Support Vector Regression |
| MCC | Matthews Correlation Coefficient |
| MFGAN | Multi-omics Fusion Graph Attention Network |
| ML | Machine Learning |
| MLR | Multiple Linear Regression |
| MOGEO | Multi-Objective Golden Eagle Optimization |
| MPL | Max-Pooling Layer |
| MSDLM | Multi-Stain Deep Learning Model |
| MTIGCN | Multi-task Interaction Graph Convolutional Network |
| MVMO | Multi-View Multi-Omics |

| Sample | Abbreviations |
| --- | --- |
| NCA | Neighborhood Component Analysis |
| NNCAE | Non-Negativity-Constrained Auto Encoder |
| NNs | Neural Networks |
| PCC | Pearson Correlation Coefficient |
| PRAD | Prostate Adenocarcinoma |
| PSO | Particle Swarm Optimization |
| ReLU | Rectified Linear Activation Function Unit |
| RF | Random Forest |
| RMSE | Root Mean Square Error |
| RNNs | Recurrent Neural Networks |
| ROC | Receiver Operating Characteristic Curve |
| RS | Rank Search |
| SCC | Spearman Correlation Coefficient |
| SVM | Support Vector Machine |
| TCGA | The Cancer Genome Atlas |
| TGDRP | Twin Graph Neural Networks For Drug Response Prediction |
| TGSA | Twin Graph Neural Networks Similarity Augmentation |
| WHO | World Health Organization |

Most studies need to specify which variants of the datasets were used, whereas the noisy dataset and the class imbalance values of the classifier will make it inadequate for learning multidimensional datasets and diminish the overall prediction accuracy. Moreover, identifying the most relevant features for predicting drug responses and handling missing values in gene expression data remains challenging. In addition to computational complexity, several approaches, especially those involving deep learning and complex metaheuristic algorithms, face high computational demands. Table 1 represents a list of all symbols and abbreviations in the article.

## METHODOLOGY

The proposed model consists of two essential stages: feature extraction and classification. At first, the dataset is preprocessed to remove noise and then employed for feature extraction. Once the preprocessing stage is done, the NNCAE model is used in the feature extraction stage. Its deep feature learning approach fixes the class imbalance problem. The obtained features are utilized for training the optimized hybrid deep learning model, Conv-LSTM, which is kept in the classification stage. This classification stage contains a hybrid model that incorporates CNN and LSTM models. Then, the hyperparameters of these models are tuned with the influence of the GEO optimization technique. In order to train the hybrid GEO-Conv-LSTM model, which predicts drug response, attributes extricated by NNCAE are last. The working flow of the proposed methodology is displayed in Fig. 1.

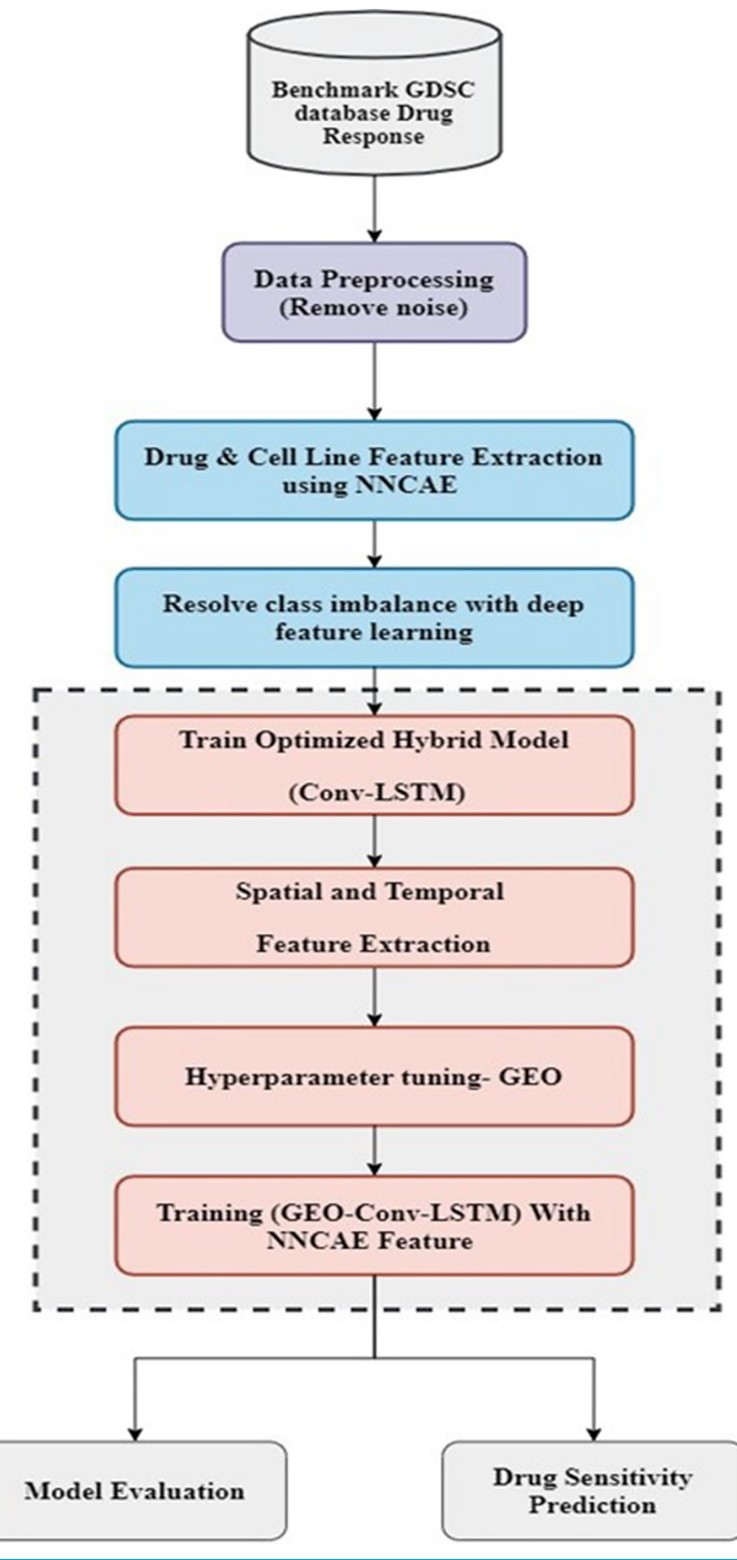

**Figure 1 The working flow of the proposed methodology.**

| Table 2 Dataset description. | | | | | | |
|---|---|---|---|---|---|---|
| Dataset | Samples | Features | Cell lines | Cell line variants | Cell line data samples | No. of Drugs | Drug data samples |
| GDSC1 | 394,241 | 22 | 1,001 | 692 | 410,125 | 403 | 333,292 |
| GDSC2 | 305,351 | 22 | 978 | 692 | 409,700 | 297 | 243,466 |

## Data description

The cancer disease drug response data and the cell lines datasets are used in this study and downloaded from the Genomics of Drug Sensitivity in Cancer (GDSC) repository (*Partin et al., 2022*) (https://www.cancerrxgene.org/downloads/). The GDSC1 and GDSC2 datasets are Pan-cancer datasets and contain twenty cancer types. The GDSC data contain the gene expression, drug data, and copy number data. The ANOVA results of these datasets have combined all the vital features, including the genetic features, cell line features, drug features, and genetic feature variants. Therefore, it can be used as the primary dataset. The GDSC1 dataset contains 394,241 gene expression data samples, 22 features, 1,001 cell lines, 692 cell line variants, and 410,125 cell line data samples created by combining different cell lines and cell line variants with the feature subsets. The GDSC2 dataset contains 305,351 gene data samples, 22 features, 978 cell lines, 692 cell line variants, and 409,700 cell line data samples. The standard drugs from the Drug Bank database are derived for drug-target information. For evaluation, the GDSC1 uses 403 drugs and 333,292 drug data samples, and GDSC2 uses 297 drugs and 243,466 drug data samples. GDSC datasets have 1,001 cell lines in GDSC1 and 978 in GDSC2. However, there are 692 cell line variants. Using these cell lines and variants, the cell line data samples are generated in the GDSC, which amounted to 410,125 in GDSC1 and 409,700 in GDSC2 datasets. These two datasets can be utilized to estimate the sensitivity of selected drugs over the cancer cell lines of the gene data. The gene data indicates the gene expression data of the cancer patients collected in GDSC after the specific drugs are given. Table 2 describes the datasets used in this article for evaluation.

## Preprocessing utilizing NNCAE

First, features/columns with more than 1/3 of the missing data are replaced through the imputation process. Then, the CDAE is introduced to learn the noisy data from the input GDSC datasets and remove them along with outlier data features. Each column is analyzed, and whichever column has more than 1/3 of the empty cells is initially considered missing data, and the entire column is imputed using the KNN imputation process. Therefore, no features or cell lines are removed at any stage. Only the missing values are imputed using KNN for better analysis of the datasets. NNCAE also identified the missing features. Then, the averaging base imputation of the KNN imputation method is used if the data is missing. If missing features cause column variations, the missing value will be replaced by the next column value. The NNCAE will determine the decision of this selection. Further, NNCAE is implemented to address the class imbalance problem.

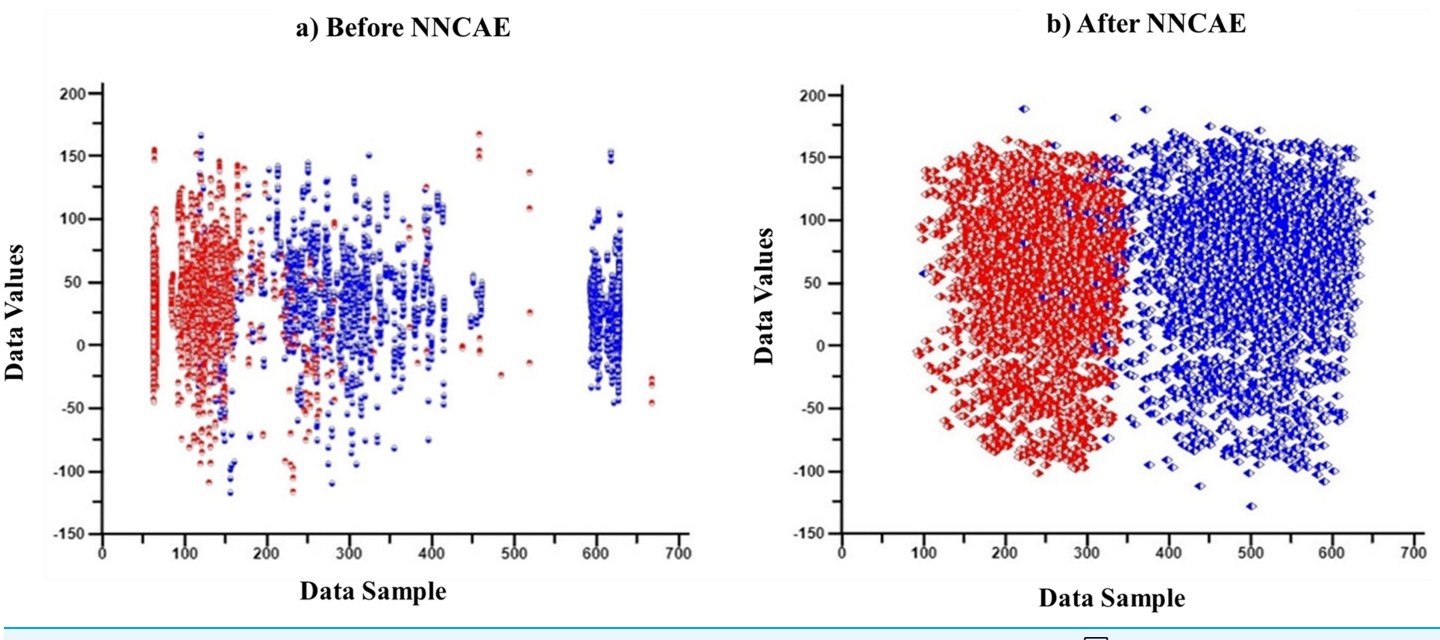

**Figure 2 The cancer dataset classification.** (A) Original dataset, (B) Preprocessing dataset.

Figure 2 shows some data points from the GDSC datasets before and after applying NNCAE for class imbalance problems.

AE is a neural network representing the data based on unsupervised learning. The objective of AE is to reconstruct the input data in the output stage. The constrained learning approach is used to extract essential information for classification, which could get the information from sparse data without any negotiation. Applying significant limitations in the network parameters forces AE to learn the hidden structures. This limitation could fix the hidden layer's size, which is given as an input. Likewise, the constraints might be nonnegativity, sparsity, and weight-decay regularization to be applied to the learned attributes. In this context, nonnegativity is imposed and forms an NNCAE (*Ayinde & Zurada, 2018*). Sparsity is essential because it facilitates an automatic and effective feature selection process. The magnitude of AE shrinks with regularization, which enhances the generalizability. Hence, constrained AEs are more beneficial for dimensionality reduction and extracting nonnegativity features. The ideology of nonnegativity is obtained from the non-negative matrix factorization, otherwise known as non-negative matrix approximation. This procedure is utilized to analyze high-dimensional data. It extricates the scattered and essential data from the collection of non-negative vectors. The nonnegativity constraint procedure aids in learning sparse and part-based data representations. The part-based data representation is executed by deconstructing the data as parts.

Reconstructing input data in AE is improved by decomposing it into scattered parts in the encoding layer. Then, it is combined in the decoding layer in an additive manner. The

non-negative mechanism is applied to the connecting weights to attain this performance improvement.

The fundamental structure of single-layered AE includes an encoder, decoder, and activation. This incorporates input, hidden, and output layers. The hidden layer encodes the input data, and the decoding operation is implemented on the output layer. The proposed model extracts the non-negativity-constrained features by following the procedure given below. The data could be represented by shattering into different distinct parts, and then adding these distinct parts must reconstruct the actual data. In this situation, the data is disintegrated in the encoding part and then reconstructed in the decoding part. The nonnegativity constraint is applied to the network work's weight for disintegration. It is a customized AE framework that can extricate non-negative features.

### $L_1$/$L_2$-non-negativity constrained autoencoder

The higher degree of nonnegativity in the network's weight is encouraged by adding a composite term to the objective function which is given in Eq. (1). This summation result is the expression of the cost function for $L_1$/$L_2$-NNCAE:

$$J_{\frac{L1}{L2}-NNCAE}(W,b) = J_{AE} + \beta \sum_{r=1}^{n'} D_{KL}\left(p \left\| \frac{1}{m}\sum_{k=1}^{m} h_j\left(X^{(k)}\right)\right.\right) + \sum_{l=1}^{2}\sum_{i=1}^{s_l}\sum_{j=1}^{s_{l+1}} fL_1/L_2\left(w_{ij}^l\right) \quad (1)$$

The weights and bias of encoding and decoding layers are represented as, $w = \{w^{(1)}, w^{(2)}\}$ and $b = \{b_x, b_h\}$, count of neurons in $l$-th layer is denoted as $s_l$, interconnection of $j$-th neuron in layer $l-1$ and $i$-th neuron in layer l for an input $X$ is denoted as $w_{ij}^l$.

$$J_{AE} = \frac{1}{m}\sum_{k=1}^{m}\left\| \sigma\left(W^{(2)}\sigma\left(W^{(1)}X^{(k)} + b_x\right) + b_h\right) - X^{(k)}\right\|_2^2 \quad (2)$$

Here, $m$ represents the count of training samples, Euclidean norm is denoted as $\|.\|_2$, Kullback–Leibler (KL) divergence for controlling the sparsity is represented as $D_{KL}(.)$, mean activations and desired activations are denoted by $p$, the count of hidden units is noted as $n'$ $h_j\left(X^{(k)}\right) = \sigma\left(W_j^{(1)}X^{(k)} + b_{x,j}\right)$ represents $j$-th hidden unit's activation function due to the input $X^{(k)}$, element-wise application of logistic sigmoid is represented as $\sigma(.)$, $\sigma(X) = \frac{1}{\exp(-x))}$, $\beta$ has the ability of controlling sparsity penalty term and

$$f_{\frac{L1}{L2}}\left(w_{ij}\right) = \begin{cases} \alpha_1 \lceil w_{ij}, k + \frac{\alpha_2}{2}\| w_{ij}\|^2 & w_{ij} < 0 \\ 0 & w_{ij} \geq 0 \end{cases} \quad (3)$$

Here, $L_1$ and $L_2$'s non-negativity-constraint weight penalty factors are specified as $\alpha_1$ and $\alpha_2$, the values of $p, \beta, \alpha_1$ and $\alpha_2$ have to be set experimentally. The weight could be upgraded with the below given error propagation formula.

$$W_{ij}^{(l)} = W_{ij}^{(l)} - \varepsilon \frac{\partial}{\partial W_{ij}^{(l)}} J_{\frac{L1}{L2}-NNCAE}(W,b) \quad (4)$$

$$b_i^{(l)} = b_i^{(l)} - \varepsilon \frac{\partial}{\partial b_i^{(l)}} J_{\frac{L_1}{L_2}} - NNCAE(W, b) \tag{5}$$

Here, $\varepsilon > 0$ represents the rate of learning and loss function of $L_1/L_2$-NNCAE is computed as follows:

$$\frac{\partial}{\partial w_{ij}^{(l)}} J_{\frac{L_1}{L_2}-NNCAE}(W, b) = \frac{\partial}{\partial w_{ij}^{(l)}} J_{AE}(w, B) + \beta D_{KL} \frac{\partial}{\partial w_{ij}^{(l)}} \left( p \| \frac{1}{m} \sum_{k=1}^{m} h_j \left( X^{(k)} \right) \right) + g \left( w_{ij}^{(l)} \right) \tag{6}$$

Here, $g(w_{ij})$ is a composite function which denotes the derivative of $f_{\frac{L1}{L2}}(w_{ij})$ given in Eq. (3) with respect to $w_{ij}$ as in Eq. (7)

$$g(w_{ij}) = \begin{cases} \alpha_1 \nabla_w \| w_{ij} \| + \alpha_2 w_{ij} & w_{ij} < 0 \\ 0 & w_{ij} \geq 0 \end{cases} \tag{7}$$

Equation (1) uses a penalty function from NNCAE, adjusting the $\alpha 1$ value to 0. The weight distribution remains nonnegative even under nonnegative constraints due to the $L2$ norm. This penalty transforms initially negative weights into positive values or close to zero, diminishing their negativity. The $L2$ norm penalty is used in machine learning. It operates by applying a penalty equal to the square of the weights' values to reduce overfitting by deterring excessively large weights, improving efficiency, and potentially increasing accuracy.

## Implication of imposing nonnegative parameters with a composite decay function

When the reconstruction error in Eq. (2) is added with the Frobenius norm of weight matrix ($\alpha \| W \|_F^2$) then it enforces Gaussian before the distribution of weight. However, the usage of the composite function in Eq. (3) resulted in the enforcement of a positively skewed deformed Gaussian distribution. The parameters $\alpha_1$ and $\alpha_2$ are used to adjust the degree of non-negativity. These parameters could be selected appropriately for imposing non-negativity and it must be ensured simultaneously to get better outcomes in supervised learning. The consequences of applying $L_1$, $L_2$ and $L1/L2$ on weight, updates are noted and observed that $L1/L2$ regularization imposes intense weight decay than $L_1$ and $L_2$. In the case of weight distribution with more positively skews leads to the reduced weight decay function. The significances of shrinking expressed in Eq. (1) are as follows: reduction of mean reconstruction error, hidden layer's activation sparsity is maximized due to the enforcement of many negative weights to 0, increase of weights with non-negative values. As a result of penalising the weights concurrently with the L1 and L2 norms, high positive connections are preserved while their orders of magnitude are narrowed. However, L1 norm given in Eq. (3) is not distinguishable at the origin which results in instability. In order to over this limitation, the smoothing function which approximates the L1 norm is

utilized. The smoothing function approximates the L1 norm for any finite dimensional vector z and the positive constant k is given as:

$$\lceil (z, k)\{\|z\| \quad \|z\| > k \quad \frac{\|z\|^2}{2k} + \frac{k}{2} \quad \|z\| \leq k \tag{8}$$

Gradient,

$$\nabla_z \lceil (z, k) = \begin{cases} \frac{z}{\|z\|} & \|z\| > k \quad \frac{z}{k} \quad \|z\| \leq k \end{cases} \tag{9}$$

For accessibility, Eq. (8) is utilized to smoothen the L1 penalty function. Both $L_1$ and $L_2$ penalized the negative weights, in which most weights are imposed as non-negative, enhancing the network's interpretability. When non-negativity is encouraged, AE enforces the layers for learning the input as a part-based representation, resulting in increased classification accuracy before fine-tuning. Hence the classification result will not degrade significantly after fine-tuning.

## GEO-Conv-LSTM

The suggested hybrid classifier is made by combining CNN and LSTM models. These models make up the Conv-LSTM module, and the GEO algorithm is used to tune its hyperparameters. In this system, the hybrid classifiers are utilized to predict cancer patients' drug responses. The drug response dataset of GDSC is usually huge and takes a lot of processing time. This limitation is overcome with the proposed Conv-LSTM. In addition, the input dataset has a problem with class imbalance due to poor data distribution. When the dataset contains more instances in one class and fewer instances in other classes, it is known as a class imbalance problem and highly deteriorates classification performance. The class imbalance problem in ML is solved by augmenting the data, but it infuses the large number of minority samples in the dataset. This leads to an overfitting problem, which can be solved using optimization techniques. With the existence of a large number of local optima, the best global solution is chosen using the GEO algorithm. The golden eagle's intelligence in hunting behaviour is used to tune the hyperparameters of the Conv-LSTM model.

This system uses spatial and temporal properties together using Conv-LSTM for predicting the drug response. The CNN has high precision and scalability and it performs better for complex problems like non-linear classification (*Habib & Qureshi, 2022*). The spatial features are extracted utilizing CNN from the gene expression data by performing different operation in different stages, which include convolutional and pooling layers. CNN uses transfer learning approach and it requires large amount of training data, therefore, it utilize the learned weights that are basically trained to handle other problems. From these learned features, some information could be used for predicting new drug response. CNN learns the spatial feature effectively but does not have the

ability to learn temporal features effectively, while the LSTM has higher capability for learning the temporal features. Hence, the LSTM model is implemented to learn the temporal features of gene expression data. The fundamental LSTM framework includes cell, input/output and forget gates (*Sherstinsky, 2020*). The three gates control the data flow inside the cell, and the cell is in charge of remembering the values over the independent intervals.

While extricating the data features, the temporal data is represented in the first dimension, and the spatial information is constituted in the second and third dimensions, and the spatial data is considered as the important features. With the usage of spatial data, the future states are predicted with inputs, and the past states by considering the nearby nodes. These predictions are done with the Hadamard product and convolutional operators (*Karapetyants & Samko, 2020*). In neural network topologies, the Hadamard product and convolutional operators play crucial roles in the processing and integration of data and features. The Hadamard product allows for the modulation and control of individual feature contributions, while convolutional operators are crucial for feature extraction and spatial analysis.

In addition, the constant errors around the nature of the nodes are protected with the Hadamard product, which makes sure that all the nodes have the ability to learn the essential features. In this model, the cell line data and drug data are given as inputs to CNN as separate branches. CNN has three layers: the convolution layer, which convolves the input feature space and then it is down-sampled by the max-pooling layer (MPL). The max pooling layer takes the highest values for flattening the output space, and the drug features are learned.

Moreover, the analysis of two branches is done with fully connected network (FCN) and the dropout probability could be fixed before FCN which limits the over-fitting problem. Further, LSTM considers the flattened dimensions of feature vector as dimension of time stamp and then uses global MPL for extricating the essential features of cell line data. The computation for switching between the states could be replaced with the convolution, this provides the ability of learning both the spatial and temporal attributes. During the construction of Conv-LSTM, the states of instances are modified in different levels and the features embedding are updated to enhance the performance of model. Finally, probability distribution over the class is generated using the activation functions such as Softmax, tanh, ReLU. In addition, generalizability is enhanced by introducing the Gaussian noise in each convolution and LSTM layers. Figure 3 shows the structure of Conv-LSTM model.

The input gate is accountable for including data to the cell state. This data could be added by following three steps. At first, the sigmoid functions are utilized for regulating the values that are to be added for the cell state. This process is common to the forget gate and it works as a filter for the information it receives. Next, attempts to create a vector that contains all of the possible values and that can be added to the cell state. This is performed with the tanh function. Finally, the regulated values are multiplied with the generated vector and this value is added to the cell state using addition operation. The input gate's outcome is represented as:

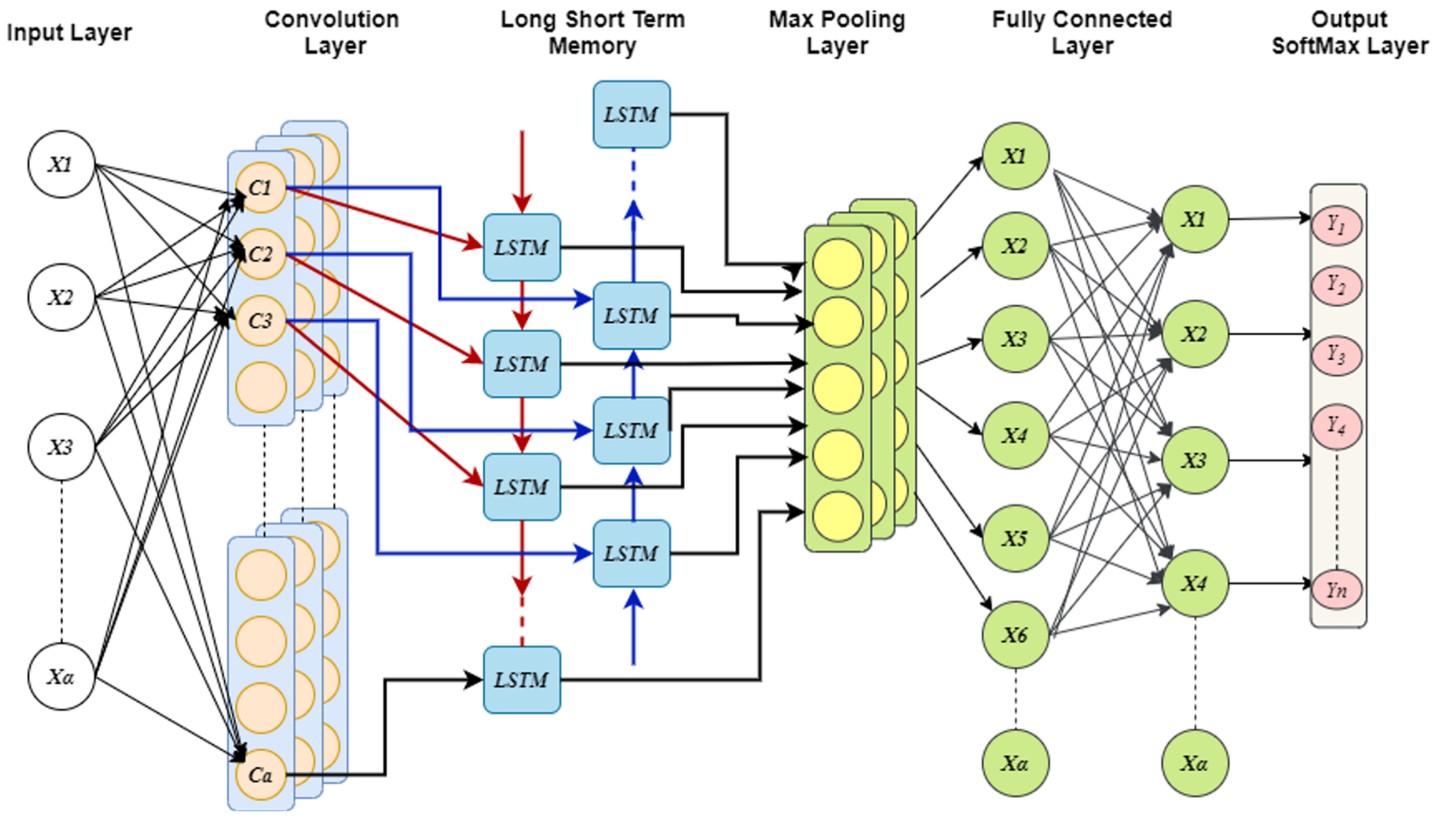

**Figure 3  Convergence graph of GEO for optimizing CONV-LSTM.**    

$$J_\alpha = \beta\left[\left(\delta_J^Y * Y_\alpha\right) + \left(\delta_J^C * C_{\alpha-1}\right) + \left(\delta_J^Z \circ Z_{\alpha-1}\right) + \eta^J\right] \tag{10}$$

where, $\beta$ indicates the gate activation function, input vector is represented as $Y_\alpha, Z_{\alpha-1}$ and $C_{\alpha-1}$ represent the previous state outcome of the memory unit and node. $\eta^J$ represents the bias of input layer with $\circ$ indicating Hadamard operator and * indicating convolutional operator. The spatial information is maintained by the convolutional operator. The weight vectors are denoted as $\delta_J^Y, \delta_J^C$ and $\delta_J^Z, \delta_J^Y$. $\delta_J^Y$ lies between the input gate and the input layer, $\delta_J^C$ is between memory outcome and the input layer, $\delta_J^Z$ between node outcome and the input layer.

As like the result of the input gate, output gate, and forget gate outputs are calculated. The representation of the forget gate outcome and the output gate outcome is given in Eqs. (11) and (12).

$$K_\alpha = \beta\left[\left(\delta_K^Y * Y_\alpha\right) + \left(\delta_K^C * C_{\alpha-1}\right) + \left(\delta_K^Z \circ Z_{\alpha-1}\right) + \eta^K\right] \tag{11}$$

The forget gate takes the input from previous cell and input of the current state and then both the inputs are multiplied with the weights. Further the bias is added after multiplication. Then the sigmoid function is given to this values and it decides what values are to be kept and discarded. The data is removed when it provides the outcome of 0 and if

it gives the outcome of 1 the information is stored for processing. The information is not required are removed by the forget gate and it optimizes the performance of LSTM.

$$L_\alpha = \beta \left[ \left( \delta_L^Y * Y_\alpha \right) + \left( \delta_L^C * C_{\alpha-1} \right) + \left( \delta_L^Z \circ Z_{\alpha-1} \right) + \eta^L \right] \tag{12}$$

where, weight vector between input layer and forget layer is represented as $\delta_K^Y$, weight vectors in memory unit and output gate is denoted as $\delta_K^C$, and weight vector between node output and output gate is represented as $\delta_K^Z$, bias related to forget gate is represented as $\eta^K$. $\delta_L^Y$ represents weight among input layer and output gate, $\delta_L^C$ indicates weight vector between memory unit and output gate, and $\delta_L^Z$ denotes weight vector among node output and output gate. Bias to output gate is denoted as $\eta^L$.

The outcome of intermediate node is computed by utilizing the activation function of the weights and it is expressed as:

$$\tilde{Z}_\alpha = tanh\ tanh\ \left[ \left( \delta_F^Y * Y_\alpha \right) + \left( \delta_F^C * C_{\alpha-1} \right) + \eta^F \right] \tag{13}$$

where, weight vector among the input layer and node is represented as $\delta_F^Y$, and $\delta_F^C$ indicates weight vector among memory unit and the node, bias of the node is represented as $\eta^F$. The bias of LSTM utilized to switch the activation function either to negative or positive side.

The node's outcome could be expressed as the addition of the transitory node state and the variance between the memory unit of past and current layers. It is given as:

$$Z_\alpha = K_\alpha \circ Z_{\alpha-1} + J_\alpha \circ \tilde{Z}_\alpha \tag{14}$$

$$Z_\alpha = K_\alpha \circ Z_{\alpha-1} + J_\alpha \circ tanh\ tanh\ \left[ \left( \delta_F^Y * Y_\alpha \right) + \left( \delta_F^C * C_{\alpha-1} \right) + \eta^F \right] \tag{15}$$

The memory unit's outcome is estimated as follows:

$$C_\alpha = L_\alpha \circ tanh\ tanh\ (Z_\alpha) \tag{16}$$

where, current outcome of memory unit is denoted by $C_\alpha$ and output gate is represented as $L_\alpha$. Hence the outcome of output layer is procured as:

$$G_\alpha = \mu \left( \delta_G^C. C_\alpha + \eta^G \right) \tag{17}$$

where, $G_\alpha$ denotes output vector of the output layer, $\delta_G^C$ represents weight vector among memory unit and output vector and bias of output layer is represented as $\eta^G$.

The outcome of convolutional layer is given as the input for LSTM and the replicas obtained from the training layers of LSTM are forwarded for drug response prediction. Therefore, the bias and the weights of the optimized Conv-LSTM can be represented as $\eta \in \{\eta^F, \eta^J, \eta^K, \eta^L\}$ and $\delta \in \{\delta_G^C,\ \delta_F^Y,\ \delta_F^C,\ \delta_L^Y,\ \delta_L^C,\ \delta_L^Z,\ \delta_K^Y,\ \delta_K^C,\ \delta_K^Z,\ \delta_J^Y,\ \delta_J^C,\ \delta_J^Z\}$. The hyper parameters, bias and weights can be tuned by the GEO to obtain optimal configuration for Conv-LSTM model.

## Hyper-parameter optimization utilizing GEO

The hyperparameters, weights, and bias of Conv-LSTM are tuned utilizing the GEO algorithm which provides optimal values. The GEO algorithm is obtained from the

hunting behavior of golden eagles (*Mohammadi-Balani et al., 2021*). The working of GEO is dependent on the golden eagle's spiral motion in the hunting process. The golden eagle has the capability of memorizing the best location it has used so far. The eagles exhibit dual motivations during their foraging behavior, characterized by the impulse to attack prey and the necessity to continue searching for optimal food sources. These concurrent desires influence this species' hunting strategy. Moreover, the golden eagle possesses the capacity to anticipate the energy dynamics of its hunting actions. Specifically, when targeting smaller prey that require a high energy expenditure, the net energy gain may not adequately compensate for the energy invested in the hunt. In the other case, if it spends more time on searching the big prey, it might catch nothing and the energy gets ruined. These two extremes are intelligently handled by the golden eagles. It catches the best prey in a reasonable amount of time and energy. They can easily transit from high attack low cruise profile to low attack high cruise profile. Every golden eagle begins the hunt by flying in large circles at high altitudes within its domain in search of prey. When prey is spotted, it begins to move around the circumference of a hypothetical circle centered on the prey. The golden eagle remembers the prey's location but it continued to circle the prey. The eagle gradually lowers its altitude while getting closer to the prey, causing the radius of the hypothetical circle around the prey to shrink. At the same time, it searches the surrounding areas for better alternatives. Golden eagles will occasionally share the location of the best prey they have found with other eagles. The hunting behaviour of the golden eagle is described in four stages. Initially, the prey is searched in spiral trajectory by the eagle and then attacks the prey by considering a straight path. In the second stage, they have more propensities in the initial and final stages of hunting. Third stage, it follows the same procedure in every flight for both attack and cruise. Finally, it takes the other eagle's information to have the prey. This hunting pattern of the golden eagle is used for hyperparameter optimization of Conv-LSTM.

The mathematical modelling of GEO algorithm is mainly based on the hunting pattern of the golden eagles. This algorithm is explained as follows:

The golden eagle uses spiral trajectory to get the prey. The golden eagle $i$, initially selects the prey of other eagles $f$ in random manner and it circle the best location of golden eagle $f$. Its own memory could be used in selecting the prey. So, $f \in \{1, 2, \ldots, PopSize\}$. The search is represented in 2D space.

At first, the prey is selected by performing cruise and attack operations. The prey location could be modelled as best location discovered by flock of golden eagles. The memory of entire flock is considered in selecting the prey for every iteration. Every search agent chooses the prey based on the memory of entire flock. For the selected prey attack and the cruise are computed. The memory is upgraded when it finds a better location. The prey selection memory is allocated to one golden eagle in this method regardless of the distance of the prey.

The best location visited by the golden eagle is guided to the population of golden eagle with the attack vector. The attack vector $(\overrightarrow{A_i})$ of golden eagle $i$ is computed as:

$$\overrightarrow{A_i} = \overrightarrow{X_f^*} - \overrightarrow{X_i} \tag{18}$$

Here, $\overrightarrow{X_f^*}$ denotes the best location visited by eagle $f$ and the present position of golden eagle ($i$) denoted by $\overrightarrow{X_i}$.

With the usage of the attack vector, the cruise vector is computed. Cruise is the golden eagle's linear speed toward the prey, indicating the exploration phase. The cruise vector with $n$ dimension is placed inside a tangent hyperplane to the circle. Hence the tangent hyperplane is computed to find the cruise vector. The scalar representation of hyperplane in n-dimensional space is expressed as:

$$h_1 x_1 + h_2 x_2 + \ldots + h_n x_n = d \;\Rightarrow\; \sum\nolimits_{j=1}^{n} h_j x_j = d \tag{19}$$

Here, $\vec{H} = [h_1, h_2, \ldots . h_n]$ represents normal vector, variable vector is represented as $X = [x_1, x_2, \ldots x_n]$, the arbitrary point on the hyperplane is denoted as $\vec{P} = [p_1, p_2, \ldots . p_n]$ and d is the dot product of $\vec{H}$ and $\vec{P}$ which is equivalent to $\sum_{j=1}^{n} h_j p_j$. When $\overrightarrow{X_i}$ as arbitrary points and $\overrightarrow{A_i}$ as normal in the hyperplane, then the cruise vector of golden eagle $i$ in iteration t is given as:

$$\sum_{j=1}^{n} a_j x_j = \sum_{j=1}^{n} a_j^t x_j^* \tag{20}$$

where the attack vector is represented as, $\overrightarrow{A_i} = [a_1, a_2, \ldots . a_n]$, the decision variable vector is denoted as $X = [x_1, x_2, \ldots . x_n]$ and the location of selected prey is given as: $X^* = [x_1^*, x_2^*, \ldots x_n^*]$.

The cruise vector is perpendicular to the attack vector and tangent to the circle. The destination point of the cruise hyperplane is expressed as:

$$\overrightarrow{C_i} = \left( C_k = \frac{d - \sum_{j \neq k} a_j}{a_k}, \;\; C_1, \; C_2 .. C_k .. C_n = random \right) \tag{21}$$

Here, $C_k$ represents $k$ th element of destination point $C$, $a_j$ denotes $j$th element of attack vector $\overrightarrow{A_i}$, $a_k$ denotes $k$ the element of attack vector, $\overrightarrow{A_i}$.

With the destination point, cruise vector for the golden eagle $i$ in $t$ th iteration is computed. The obtained elements of destination points are lies between 0 and 1. The golden eagle's population is attracted by cruise vector than area in its memory and it plays important role in exploration phase.

The golden eagle moves to the new position depending on the cruise and attack vectors. Hence, the golden eagle j's step vector in iteration t is expressed as below:

$$\Delta_{xj} = \overrightarrow{r_1} P_a \frac{\overrightarrow{A_j}}{\|\overrightarrow{A_j}\|} + \overrightarrow{r_2} P_c \frac{\overrightarrow{C_j}}{\|\overrightarrow{C_j}\|} \tag{22}$$

where, $P_a^t$ denotes the attack coefficient at iteration t, cruise coefficient at iteration t is denoted as $P_c^t$. Random vectors are represented as $\overrightarrow{r_1}$ and $\overrightarrow{r_2}$ which lies between [0 and 1]. $\|\overrightarrow{C_j}\|$ and $\|\overrightarrow{A_j}\|$ denotes the Euclidean norm of cruise and attack vectors. These two factors are computed as follows:

$$\|\vec{C_j}\| = \sqrt{\sum_{j=1}^{n} c_j^2}, \qquad \|\vec{A_j}\| = \sqrt{\sum_{j=1}^{n} a_j^2} \tag{23}$$

The step vector governs how cruise and attack affect the golden eagles. The golden eagle's new position is expressed as:

$$x_j^{t+1} = x_j^t + \Delta_{xj}^t \tag{24}$$

The memory would be updated to the new position when the fitness function $j$ gives better outcome than the previous position.

In order to shift from the exploration state to the exploitation state, the GEO algorithm utilizes cruise and attack coefficients $(P_a, P_b)$ respectively. These two coefficients are calculated by the linear expressions given as:

$$\left\{ P_a = P_a^0 + \frac{t}{T} \left| P_a^T - P_a^0 \right| \quad P_c = P_c^0 + \frac{t}{T} \left| P_c^T - P_c^0 \right| \right. \tag{25}$$

The initial values of propensity to the cruise and attacks are represented as $P_c^0$ and $P_a^0$, the present iteration is denoted as $t$, and the maximum number of iterations is given as $T$, The resultant values of propensity to cruise and attack are denoted as $P_c^T$ and $P_a^T$.

The multi-objective problems have multiple objective functions, which generates difficulty in the optimization mechanism, which is not considered in the single objective problems. A typical multi-objective problem is expressed as:

$$\text{Minimize } F(\vec{x}) = \{f_1(x), f_2(x), \dots f_n(x)\} \tag{26}$$

Subject to:

$$g_i(\vec{x}) \leq 0, \quad i = 1, 2, 3 \dots r$$
$$h_i(\vec{x}) = 0, \quad i = r+1, r+2, \dots s$$

where F represents the collection of objectives needs to be optimized, $\vec{x}$ denotes the design/decision variables, $i$-th inequality constraint is represented as $g_i(\vec{x})$, $i$-th equality constraint is denoted as $h_i(\vec{x})$ and the total number of constraints and number of inequality constraints are denoted as s and r, respectively.

The multi-objective GEO is constructed on the concept of single objective optimization. In addition to the procedures of single objective GEO, it has three concepts, which include external archive, prey prioritization criteria, and multi-objective prey selection. The concept called Pareto dominance is used to deal with multi-objective problems. The Pareto optimal solutions are used to determine the multi-objective problems. Here crowding distance could be used as density index and the crowding solution is expressed as:

$$C_i = \frac{1}{n} \sum_{j \in J} \frac{(f_{i+1,j} - f_{i,j}) - (f_{i,j} - f_{i-1,j})}{f_j^{max} - f_j^{min}} \tag{27}$$

| **Algorithm 1** Multi-Objective GEO for optimizing Conv-LSTM. | |
|---|---|

**Input:** Possible parameter values of Conv-LSTM

**Output:** Optimal Conv-LSTM configuration

1:      Initialize the population of golden eagles

2:      Assign the solutions (Conv-LSTM parameter values) to golden eagles

3:      Set the search agents

4:      Formulate objection/fitness functions $f_{i-1,j}$, $f_{i,j}$, $f_{i+1,j}$ using Eq. (26)

5:      **For** each golden eagle

6:          Compute the fitness values for each golden eagle

7:          Rank the eagles based on the fitness values

8:          Save the ranked solutions to the archive

9:      **End for**

10:     Initialize population memory

11:     Initialize $P_a$ and $P_c$

12:     **For** each iteration $t$

13:          Update $P_a$ and $P_c$ using Eq. (22)

14:          Compute crowding distance for current archive members

15:          **For** each golden eagle $i$

16:              Apply Prey selection using Roulette wheel weighted by crowding distances

17:              Assign the search agents towards the selected prey (possible solution)

18:              Calculate attack vector $\overrightarrow{A_i}$ using Equation (18)

19:              **If** the attack vector's length = 0

20:                  Return the selected prey as the best solution

21:              **Else**

22:                  Calculate cruise vector $\overrightarrow{C_i}$ using Equations (19–21)

23:                  Calculate step vector $\Delta_{xj}$ using Equations (22) and (23)

24:                  Update position using Equation (24)

25:                  Evaluate fitness values for the new position

26:                      **If** the new position is non-dominated to the current archive members

27:                      **If** the external archive is not full

28:                          Add the new solution to the archive

29:                      **Else**

30:                          Calculate the sparsity distances using Equations (27) and (28)

31:                          **For** each archive member

32:                              Apply roulette wheel weighted by sparsity distances for eliminating archive member

33:                              Replace the eliminated solution with the new one

34:                          **End for**

35:                      **End if**

| **Algorithm 1** (continued) | |
|---|---|
| 36: | **End if** |
| 37: | **End if** |
| 38: | Return the final best solution as the optimal configuration of Conv-LSTM |
| 39: | **End for** |
| 40: | **End for** |

For every archive member, the crowding distance is computed. When the archive is sorted by the objective values of the j-th objective function, there are three consecutive members such as $f_{i-1,j}$, $f_{i,j}$, $f_{i+1,j}$. Sparsity scores $S_i$ are the new scores used for the roulette wheel procedure. It could be computed as:

$$S_i = 1 - C_i \tag{28}$$

The prey selection criterion is varied that of the single GEO. The search agents use their own memory for prey selection in a single GEO, but in the case of multi-objective GEO (MOGEO), the archive uses the non-dominated location it visited so far. The roulette wheel concept is used in the prey selection of MOGEO. In this procedure, sparsity scores of the current archive members are used to calculate the weights. Based on the mathematical modelling of GEO, the implementation of multi-objective problems is given in Algorithm 1.

The procedures followed in GEO are utilized to tune hyperparameters including weights and bias of Conv-LSTM. This tuning procedure provides optimal configuration for the Conv-LSTM model. The Conv-LSTM fitness function $f(y)$ is replaced by a function $f(\delta, \eta)$ and it is related to the layers of CNN, activation function, and learning rate of LSTM.

## RESULTS AND DISCUSSION

The proposed NNCAE and GEO-Conv-LSTM-based drug response prediction model is evaluated using experiments conducted over the GDSC1 and GDSC2 datasets for 20 cancers. The experiments are performed using the DL toolbox of MATLAB R2021a on a computer with configurations of an i7 processor, 8GB RAM, 1TB hard disk, 512GB SSD, and 4GB GEFORCE RTX 3050 (NVIDIA, Santa Clara, CA, USA) GPU.

In order to maintain spatial and temporal feature extraction, Conv-LSTM is proposed. The input gene expression data is given to the CNN model and the output is obtained in the LSTM model, which uses the input from CNN. This generates sequences at every time step. Further, the sequence vector is applied to the fully connected layer before the softmax layer for probability distribution over the classes. The optimization of parameters is done by utilizing the GEO algorithm. Figure 4 depicts the convergence curve that GEO obtained when optimizing the Conv-LSTM's parameters.

The GEO procedures are utilized to tune hyperparameters, including weights and bias of Conv-LSTM. This tuning procedure provides optimal configuration for the Conv-LSTM model. The Conv-LSTM fitness function f(y) is replaced by a function f(δ,η), and it is

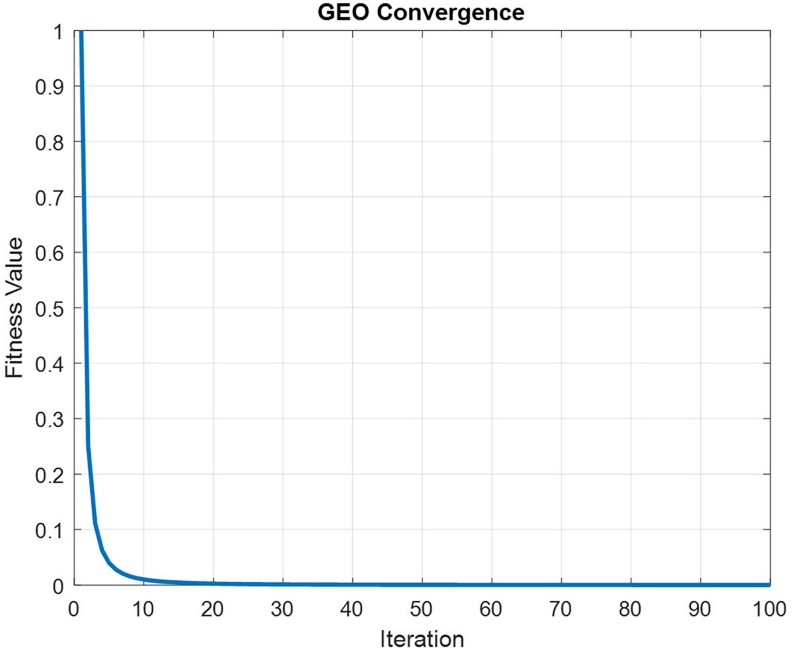

**Figure 4 The schematic of a CONV-LSTM model.**

**Table 3 Optimal parameters of conv-LSTM obtained using GEO.**

| Layer | Name | Baseline | GEO optimized |
|---|---|---|---|
| 0 | Input | – | (1, 256, 24) |
| 1 | Convolution Pooling Activation | (16, 3, 3) (2, 2) Sigmoid, ReLU | (16, 256, 24) (16, 128, 12) Sigmoid |
| 2 | Convolution Pooling Activation | (32, 3, 3) (2, 2) Sigmoid, ReLU | (32, 128, 12) (32, 64, 6) Sigmoid |
| 3 | Convolution Pooling Activation | (64, 3, 3) (2, 2) Sigmoid, ReLU | (64, 64, 6) (64, 32, 3) Sigmoid |
| 4 | Convolution Pooling Activation | (128, 3, 3) (2, 2) Sigmoid, ReLU | (128, 32, 3) (128, 16, 1) Sigmoid |
| 5 | Convolution Pooling Activation | (256, 3, 3) (2, 2) Sigmoid, ReLU | (256, 16, 1) (256, 8, 1) Sigmoid |
| 6 | Flatten | – | (1536, ) |
| 7 | Fully connected | 278 | (278, ) |
| 8 | LSTM Activation Learning rate | 200 tanh, ReLU 0.5 | 128 tanh 0.1 |
| 9 | LSTM Activation Learning rate | 200 tanh, ReLU 0.5 | 128 tanh 0.1 |
| 10 | Fully connected | 278 | (278, ) |

related to the layers of CNN, the activation function, and the learning rate of LSTM. Table 3 shows the optimal parameters of Conv-LSTM obtained using the GEO algorithm.

Considering the configurations from the above table, the Conv-LSTM configuration is formulated and used in the classification process for drug sensitivity prediction.

The proposed model is implemented along with the relevant existing methods from the literature. The comparisons are made in terms of accuracy, precision, recall, f-measure, specificity, root mean square error (RMSE), Spearman correlation coefficient (SCC), Pearson correlation coefficient (PCC) and processing time.
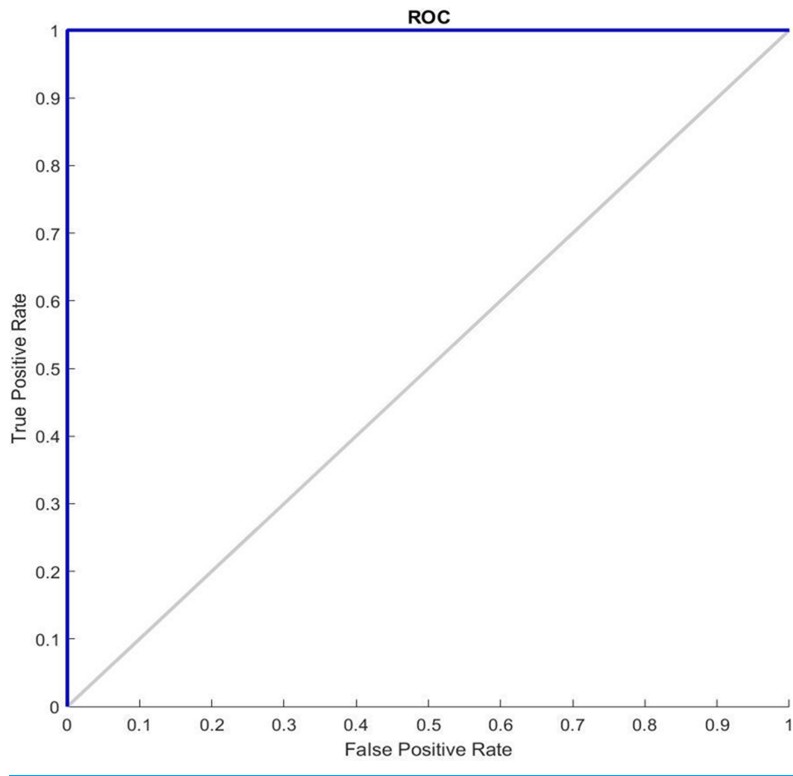

**Figure 5** ROC Curve of GDSC1 data.               

Figure 5 shows the ROC graphs obtained for the GDSC1 datasets. In a typical ROC curve for GDSC1 classification, a plot of the true positive rate is observed against the false positive rate as the discrimination threshold varies. The curve illustrates how well the classifier can distinguish between malignant and non-malignant samples in the GDSC1 dataset. A classifier with good performance will have a curve that rises steeply and approaches the top-left corner of the plot. The ROC value quantifies the overall performance of the classifier. A high ROC (close to 1) indicates excellent discrimination ability, while an ROC of 0.5 indicates average performance. It assesses the performance of predictive models for distinguishing between cancerous and non-cancerous breast tissue samples. Higher AUC values suggest better model performance in distinguishing between the two classes. ROC analysis for the GDSC2 dataset would follow a similar pattern to that of the GDSC1 dataset, assessing the performance of classifiers in distinguishing between cancerous and non-cancerous prostate tissue samples.

Table 4 shows the results obtained by the proposed NNCAE-GEO-Conv-LSTM for the GDSC1 datasets compared against the state-of-the-art Conv-LSTM method and the proposed GEO-Conv-LSTM.

From Table 3, it is evident that the proposed NNCAE and GEO-Conv-LSTM-based DRP model has a higher prediction performance of breast cancer. The proposed approach has achieved high values of accuracy, recall, f-measure, SCC, and PCC and reduced RMSE error rate. The proposed NNCAE-GEO-Conv-LSTM has achieved 8.12% and 9% better

**Table 4 Performance evaluation on GDSC1.**

| Parameters/Methods | Conv-LSTM | GEO-Conv-LSTM | NNCAE-GEO-Conv-LSTM |
|---|---|---|---|
| Accuracy | 0.9081 | 0.9298 | 0.9699 |
| Precision | 0.8432 | 0.8689 | 0.9125 |
| Recall | 0.8125 | 0.8415 | 0.9286 |
| F-measure | 0.8273 | 0.8549 | 0.9204 |
| Specificity | 0.8760 | 0.9133 | 0.9351 |
| RMSE | 0.2876 | 0.2465 | 0.2236 |
| SCC | 0.7118 | 0.7074 | 0.8516 |
| PCC | 0.7037 | 0.6981 | 0.8493 |

**Figure 6 SCC and PCC for GDSC1.**

accuracy than GEO-Conv-LSTM and 11.18% and 13.44% higher accuracy than Conv-LSTM for GDSC1 and GDSC2 datasets. Figure 6 shows the plots obtained for SCC and PCC for GDSC1.

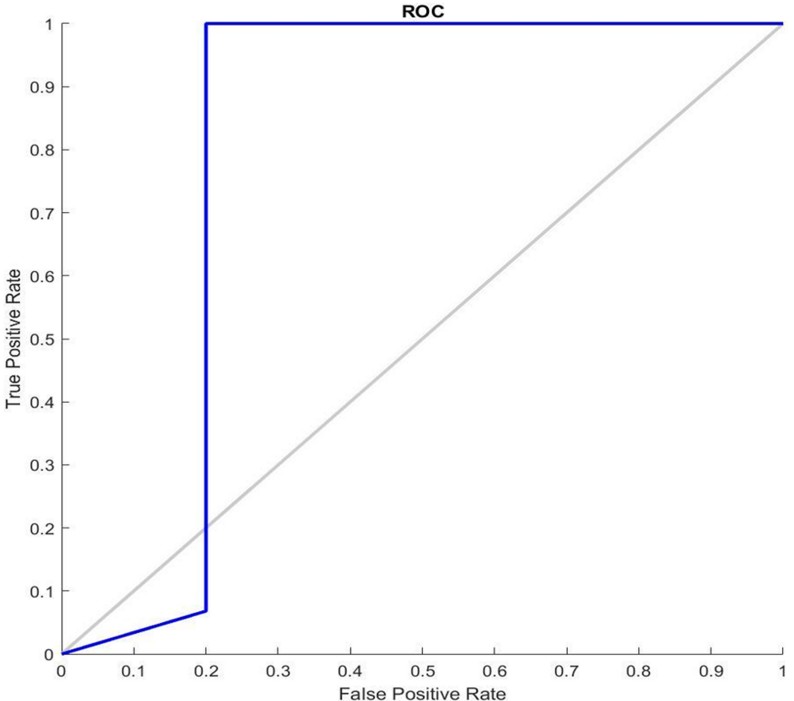

**Figure 7  ROC Curve of GDSC2 datasets.**

**Table 5  Performance evaluation on GDSC2.**

| Parameters/Methods | Conv-LSTM | GEO-Conv-LSTM | NNCAE-GEO-Conv-LSTM |
|---|---|---|---|
| Accuracy | 0.9060 | 0.9367 | 0.9778 |
| Precision | 0.8222 | 0.9064 | 0.9371 |
| Recall | 0.8439 | 0.8865 | 0.9276 |
| F-measure | 0.8329 | 0.8963 | 0.9323 |
| Specificity | 0.7981 | 0.8145 | 0.8333 |
| RMSE | 0.4883 | 0.4432 | 0.3953 |
| SCC | 0.7175 | 0.7338 | 0.8752 |
| PCC | 0.7356 | 0.7766 | 0.8953 |

Figure 7 shows the receiver operating characteristic curve ROC graphs and the best training performance obtained from the GDSC1 and GDSC2 datasets for GDSC2 diseas. ROC analysis for the GDSC2 dataset would follow a similar pattern to that of the GDSC1 dataset, assessing the performance of classifiers in distinguishing between cancerous and non-cancerous prostate tissue samples.

Table 5 shows the results obtained by the proposed NNCAE-GEO-Conv-LSTM for the GDSC2 datasets compared against the state-of-the-art Conv-LSTM method and the proposed GEO-Conv-LSTM.

Table 5 shows that the NNCAE and GEO-Conv-LSTM model-based anti-cancer DRP have a higher prediction performance of GDSC2, similar to that of GDSC1. The proposed

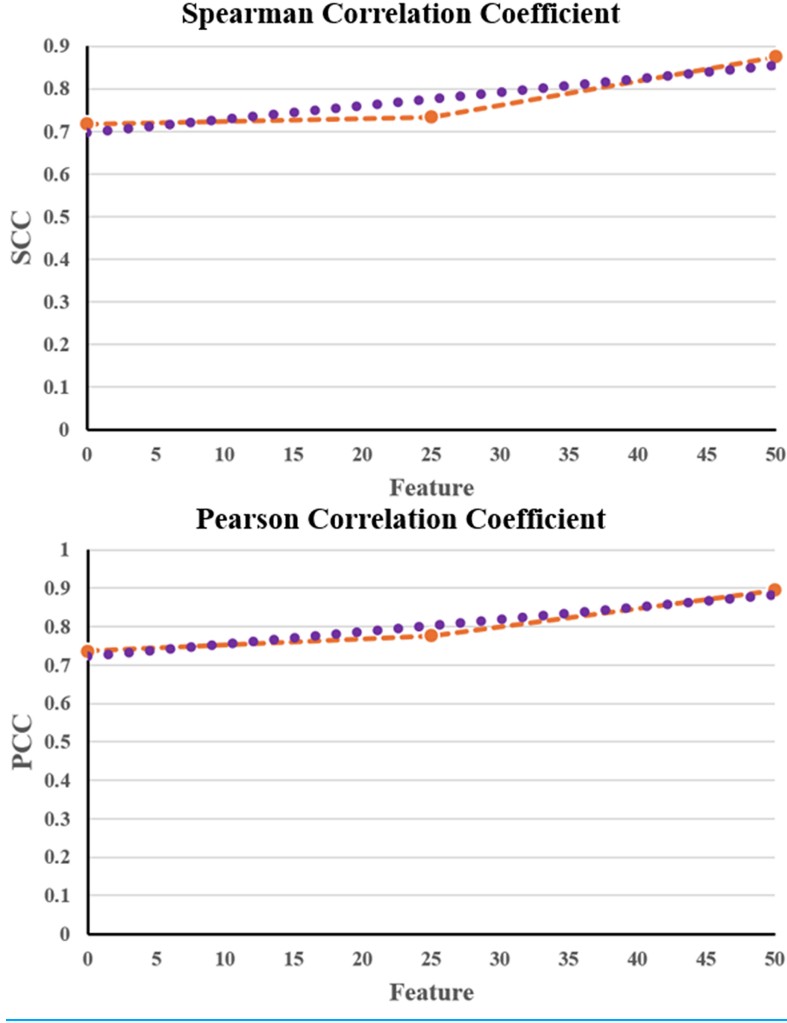

**Figure 8  SCC and PCC for GDSC2.**               

approach has achieved higher f-measure, accuracy, recall, SCC and PCC values and reduced RMSE error rates for both the GDSC1 and GDSC2. Like the GDSC1, the proposed approach has achieved competent performance for the GDSC2 dataset. The proposed NNCAE-GEO-Conv-LSTM has gained 7.18% and 4.11% better accuracy than Conv-LSTM and GEO-Conv-LSTM for the GDSC2 dataset. Figure 8 shows the plots obtained for SCC and PCC for GDSC2 datasets.

Figure 9 shows the accuracy, precision, recall, and f-measure comparison of the proposed NNCAE-GEO-Conv-LSTM and existing models.

The NNCAE-GEO-Conv-LSTM model obtains an accuracy of 0.9699, which is 6.18% and 4.01% higher than the Conv-LSTM and GEO-Conv-LSTM models. It has also obtained a precision of 0.9125, which is 6.93% and 4.36% higher than the Conv-LSTM and GEO-Conv-LSTM models. The model attained a recall of 0.9286, which is 11.61% and 8.71% higher, and the F-measure of 0.9204, which is 9.31% and 6.55% higher than Conv-LSTM and GEO-Conv-LSTM models for GDSC1 dataset. For the GDSC2 dataset, the

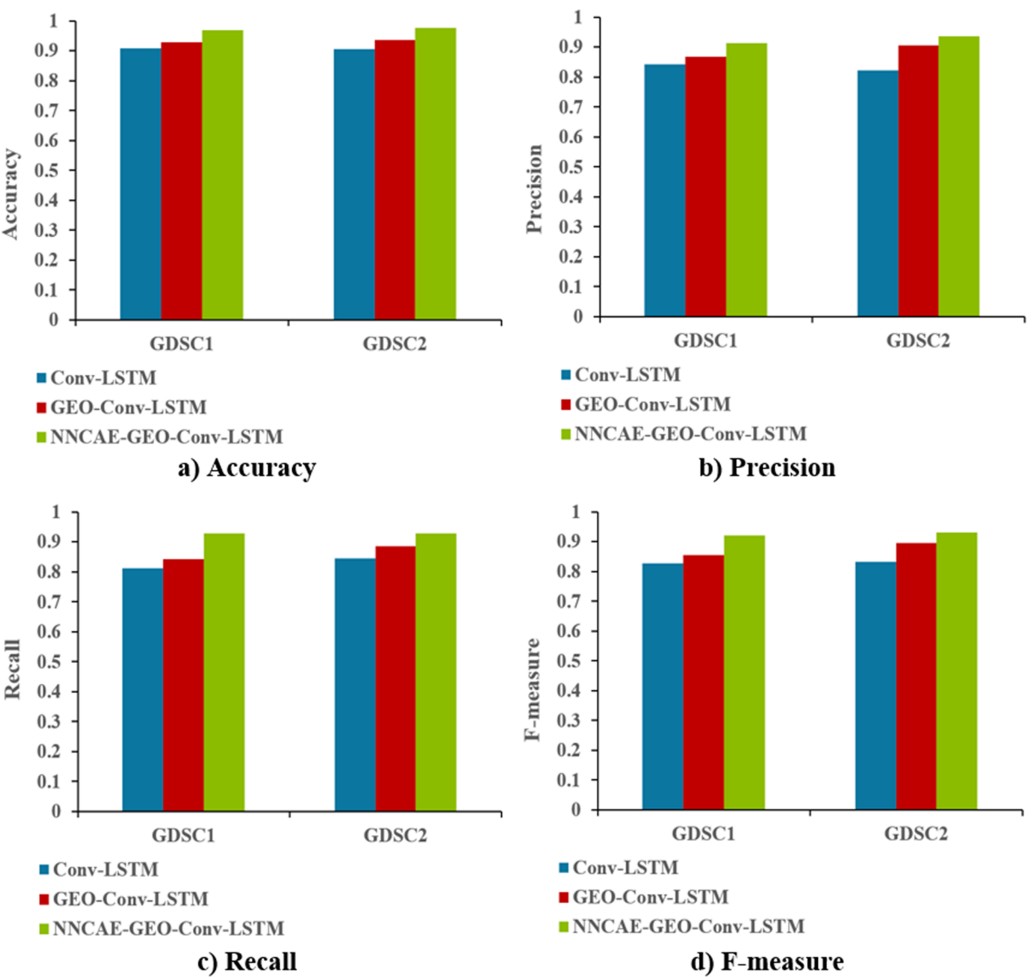

**Figure 9 Accuracy, precision, recall and F-measure comparisons.**

model attained an accuracy of 0.9778 which is 7.18% and 4.11% higher; precision obtained is 0.9371, which is 11.49% and 3.07% higher; recall obtained is 0.9276 which is 8.37% and 4.11% higher and F-measure obtained is 0.9323 which is 9.94% and 3.60% higher than Conv-LSTM and GEO-Conv-LSTM models.

Figure 10 shows the Specificity, RMSE, SCC and PCC comparisons of the proposed NNCAE-GEO-Conv-LSTM and the existing models for both GDSC1 and GDSC2 datasets.

The NNCAE-GEO-Conv-LSTM model obtains a specificity of 0.9351, which is 5.91% and 2.18% higher than the Conv-LSTM and GEO-Conv-LSTM models. It has also obtained an RMSE of 0.2236, which is 6.40% and 2.29% less than the Conv-LSTM and GEO-Conv-LSTM models. The model attained an SCC of 0.8516, which is 13.98% and 14.42% higher, and the PCC obtained is 0.8493, which is 14.56% and 15.12% higher than Conv-LSTM and GEO-Conv-LSTM models for GDSC1 dataset. For the GDSC2 dataset, the model attained a specificity of 0.8333, which is 3.52% and 1.88% higher; the RMSE obtained is 0.3953, which is 9.30% and 4.79% less. SCC obtained is 0.8752, which is 15.77%

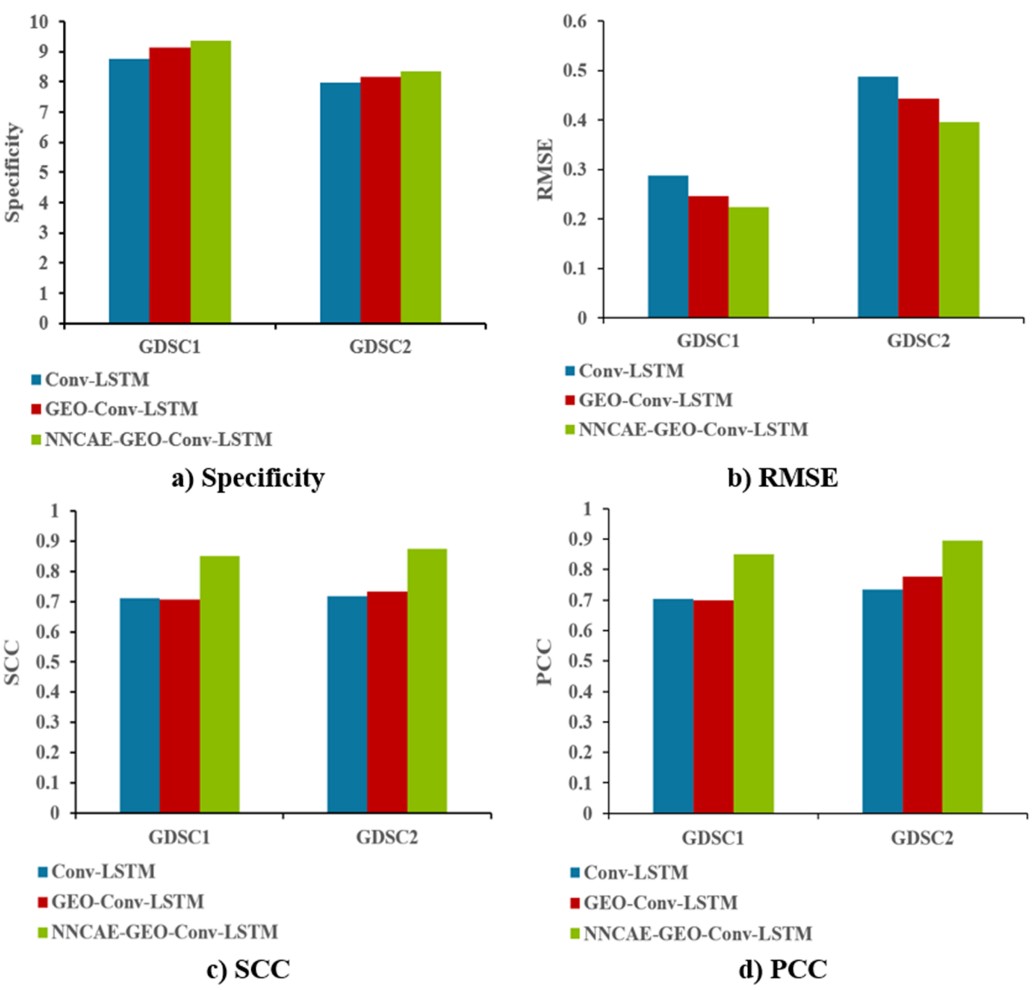

**Figure 10 Specificity, RMSE, SCC, and PCC comparisons.**

**Table 6 Processing time (in minutes) comparison.**

| Dataset | Conv-LSTM | GEO-Conv-LSTM | NNCAE and GEO-Conv-LSTM |
|---------|-----------|---------------|--------------------------|
| GDSC1 | 55.5683 | 49.4614 | 42.1611 |
| GDSC2 | 54.4671 | 50.7605 | 44.0573 |

and 14.14% higher, and PCC obtained is 0.8953, which is 15.97% and 11.87% higher than Conv-LSTM and GEO-Conv-LSTM models.

Table 6 illustrates the processing time comparison of the proposed approach for the GDSC1 and GDSC2 datasets.

Figure 11 shows the Processing time (complexity) comparisons of the proposed NNCAE-GEO-Conv-LSTM and the existing models for both datasets.

Figure 11 illustrates that the NNCAE-GEO-Conv-LSTM approach has consumed less processing time (in minutes) and reduced computational complexity. For GDSC1, the proposed approach has obtained 42.1611s, which is 24.12% and 14.75% less processing

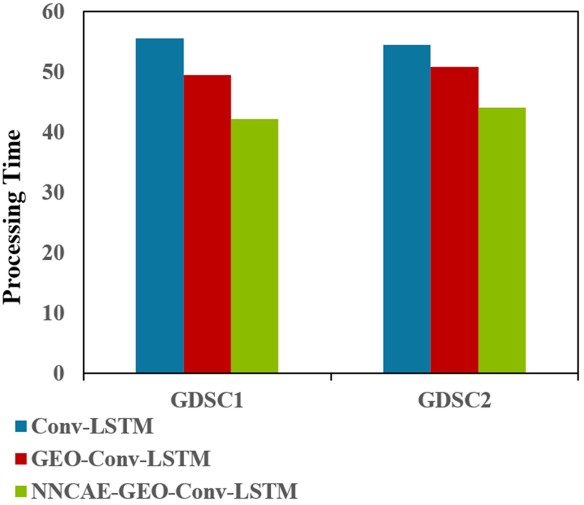

**Figure 11 Processing time (complexity) comparison.**

**Table 7 Performance comparison against literature methods.**

| Methods | GDSC1 | GDSC2 |
|---|---|---|
| CNN | 0.8887 | 0.8734 |
| SVM | 0.8943 | 0.9022 |
| SNN | 0.8654 | 0.9243 |
| RF | 0.9476 | 0.9032 |
| LSTM | 0.9501 | 0.9486 |
| Conv-LSTM | 0.9259 | 0.9566 |
| NNCAE and GEO-Conv-LSTM | 0.9699 | 0.9778 |

time than the Conv-LSTM and GEO-Conv-LSTM models, respectively. Similarly, for GDSC2, the proposed approach has obtained 44.0573, 19.11%, and 13.20% less time than the Conv-LSTM and GEO-Conv-LSTM models, respectively. Table 7 shows the performance comparison of accuracy for the model NNCAE and GEO-Conv-LSTM-based anti-cancer DRP model against the prominent methods from the literature.

As highlighted in Table 7, the proposed approach of NNCAE and GEO-Conv-LSTM has achieved high accuracy. This enhancement can be attributed to the optimal tuning of the Conv-LSTM hyper-parameters by the GEO and the data quality enhancement using NNCAE. For both the GDSC1 and GDSC2 datasets, the proposed approach has obtained an accuracy of 0.9699 and 0.9778, respectively. Therefore, maximizing the size of the GSDC dataset for model training has significantly increased the prediction accuracy.

## CONCLUSION

This work aimed to develop an efficient DRP model for GDSC1 and GDSC2 based on advanced DL classifiers. Additionally, the negative impact of the noisy, outlier data, which was imbalanced, must be tackled effectively. NNCAE and GEO-Conv-LSTM models were

developed to achieve these objectives and enhance anti-cancer drug sensitivity prediction accuracy to the cancer cell lines obtained from gene expression data. Experiments were performed on GDSC1 and GDSC2 datasets to evaluate the proposed approach for the DRP problem. The proposed NNCAE and GEO-Conv-LSTM-based DRP model obtained 96.99% and 97.79% accuracy for GDSC1 and GDSC2 data, respectively. It has also reduced the overall complexity by minimizing the processing time for both datasets. Although competent results were obtained, the proposed approach performance could also be improved by investigating the impact of data sparsity and different feature dimensions of the multimodal datasets in the future. In addition, different deep learning models will be incorporated to avoid class imbalance and model complexity, and the study will use different cancer datasets. The source code and dataset of the NNCAE-GEO-Conv-LSTM method are available at https://github.com/WesamHajim/Conv-LSTM-GEO.

### Funding
This work was supported by the Fundamental Research Grant Scheme, grant number FRGS/1/2022/ICT02/UKM/02/7, funded by the Ministry of Higher Education (MOHE) Malaysia. The funders had no role in study design, data collection and analysis, decision to publish, or preparation of the manuscript.

### Grant Disclosures
The following grant information was disclosed by the authors:
Ministry of Higher Education (MOHE) Malaysia: FRGS/1/2022/ICT02/UKM/02/7.

### Competing Interests
The authors declare that they have no competing interests.

### Author Contributions
- Wesam Ibrahim Hajim conceived and designed the experiments, performed the experiments, analyzed the data, performed the computation work, prepared figures and/or tables, authored or reviewed drafts of the article, and approved the final draft.
- Suhaila Zainudin conceived and designed the experiments, performed the experiments, analyzed the data, prepared figures and/or tables, authored or reviewed drafts of the article, and approved the final draft.
- Kauthar Mohd Daud conceived and designed the experiments, performed the experiments, analyzed the data, prepared figures and/or tables, authored or reviewed drafts of the article, and approved the final draft.
- Khattab Alheeti analyzed the data, prepared figures and/or tables, authored or reviewed drafts of the article, and approved the final draft.

## Data Availability

The source code and dataset of the NNCAE-GEO-Conv-LSTM method are available at GitHub and Zenodo:

- https://github.com/WesamHajim/Conv-LSTM-GEO

- Hajim, W. (2024). Golden Eagle Optimization CONV-LSTM and Non-Negativity-constrained Autoencoder-Deep Learning Optimization. https://doi.org/10.5281/zenodo.13997533

The data is available at Genomics of Drug Sensitivity in Cancer:

https://www.cancerrxgene.org/downloads/bulk_download. Search terms: GDSC1-ANOVA, GDSC2-ANOVA

## Supplemental Information

Supplemental information for this article can be found online at http://dx.doi.org/10.7717/peerj-cs.2520#supplemental-information.

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
