# Peer review of "Golden eagle optimized CONV-LSTM and non-negativity-constrained autoencoder to support spatial and temporal features in cancer drug response prediction"

_PeerJ Computer Science, doi:10.7717/peerj-cs.2520_

## Round 0.1 · original submission · Major Revisions

Dear authors,

Thank you for submitting your manuscript titled "Golden Eagle Optimization CONV-LSTM and Non-Negativity-Constrained Autoencoder to Support Spatial and Temporal Features in Cancer Drug Response Prediction" to PeerJ Computer Science. The manuscript has been reviewed by several experts in the field, and their feedback has been compiled.

After careful consideration, it has been determined that your manuscript requires major revisions before it can be considered for publication. While the work presents an innovative approach with potential impact on cancer drug response prediction, the reviewers have identified several significant areas that need improvement.

Summary of Required Revisions:

1. Clarity and Structure:
o The manuscript needs substantial improvements in clarity and structure. There are numerous issues with grammar, writing style, and the use of abbreviations, which make the text difficult to follow. Reviewers have strongly recommended that you have the manuscript professionally proofread and edited for language and clarity.
o The Methodology section is currently too long and includes results that should be moved to the Results section. This restructuring will help to balance the sections and make the manuscript easier to navigate.

2. Experimental Design and Methodology:
o Reviewers have raised concerns about the lack of clear explanation regarding the experimental design and data preprocessing steps. You need to provide detailed descriptions of the datasets, preprocessing methods, and the rationale behind the choices made, particularly regarding the handling of missing data and the selection of features.
o The manuscript should include a thorough discussion of the test dataset to evaluate the generalizability of the proposed model. This will also help in identifying potential overfitting, which is a common issue in deep learning applications.
3. Use of Abbreviations:
o There are many undefined and inconsistent abbreviations throughout the manuscript. It is essential to create a table of abbreviations and ensure consistent usage throughout the text to avoid confusion among readers.
4. Figures and Tables:
o The quality of the figures is inadequate, with some being blurry and lacking necessary labels. You should improve the resolution of all figures and ensure that all axes and data points are clearly labeled.
o Table 1 in particular has been noted as unclear, with ambiguous definitions for column headers such as "Samples" and "Features." You need to clearly define these terms and provide accurate counts for the distinct cell lines used.
5. Comparative Analysis and Benchmarking:
o The manuscript lacks benchmarking against alternative models. To strengthen your work, you should include comparisons with state-of-the-art models, discussing how your approach performs relative to these benchmarks.
o Additionally, you should provide more justification for the use of the Golden Eagle Optimization algorithm compared to more traditional methods like grid search or random search.
6. Literature Review:
o The literature review should be updated to include more recent and relevant studies. Reviewers have suggested that you distinguish between different methodologies in the field and cite more up-to-date sources to contextualize your research within the broader scientific landscape.
7. Technical and Conceptual Issues:
o Reviewers have identified specific technical issues, such as the explanation of "gene expression temporal and spatial features" and the unclear description of data modalities. These need to be addressed with precise definitions and explanations to avoid any ambiguity.

Next Steps:
Please address the above issues and revise your manuscript accordingly. Along with your revised manuscript, include a detailed response letter that outlines the changes made and provides justifications for any areas where changes were not made. Given the extent of the revisions required, we understand this may take some time, but these improvements are crucial for the manuscript to be considered for publication.

We look forward to receiving your revised submission and appreciate your continued interest in PeerJ Computer Science.


Best regards,

Reviewer 1 ·

Basic reporting

• The manuscript proposes and discusses an innovative deep learning architecture to improve cancer drug response prediction. Moreover, hyperparameters pertaining to the developed algorithm are tuned through a metaheuristic strategy to enhance the model's performance.
• The conceptualized idea is interesting and innovative and has brought state-of-the-art results for cancer-related data classification.
• The manuscript is generally well-written in terms of quality English. However, there are still many spelling mistakes, improper writing, and especially a lack of attention to abbreviations, which should be reconsidered.
• The literature is comprehensively reviewed, and updated studies in the specialized field are covered.
• The figures and tables presented are of good quality and resolution.
• The manuscript is not well structured. Within the Methodology Section, some results are reported (e.g., results derived from hyperparameter tuning), which should be transferred to the Results Section. This issue has caused the Methodology Section to be very long and the Results and Discussion Section to be shortened.
• The raw data used are publicly available. Moreover, the obtained results are stored in the Supplementary Materials Section.
• The obtained results are reported in a transparent and comprehensible manner.

Experimental design

• The existing challenges of traditional methods for cancer drug response prediction have been described well, and relevant ideas have been conceptualized to address them.
• The designed experiment was applied to the standard data, and the relevant results were obtained.
• Methods are described with sufficient information to be reproducible by other investigators.

Validity of the findings

• The performance of the proposed model has been evaluated through ROC analysis and diagnostic statistics thereof. The reported results indicate that the proposed model has been able to significantly improve the performance of traditional approaches in cancer drug response prediction.
• The conclusions have been appropriately stated and connected to the research questions.

Additional comments

GENERAL COMMENTS

1. I suggest that the title of the manuscript be slightly modified to reflect better the methodology adopted:
Golden eagle optimization-based CONV-LSTM ….
OR
Golden eagle optimized CONV-LSTM….

2. As a critical issue, have you considered a test dataset (in addition to the training dataset) to check the generalizability of the results? It is necessary to consider the test dataset to ensure the adequate generalizability of the proposed model. Moreover, it helps track the possibility of overfitting, a dominant phenomenon in deep/machine learning applications.

3. The authors employ a metaheuristic algorithm (i.e., golden eagle optimization) to tune the hyperparameters of the deep learning model. What is the advantage of this strategy to straightforward computational approaches such as gris search and random search?

4. The authors denoise the input data in the preprocessing step. However, feature extraction has been proven to be more efficient when the data are corrupted with noise. For example, in denoising autoencoders, we manually add noise to the data so that the training network can more effectively learn the deep representations of the input data in the bottleneck layer.

5. The manuscript is full of undefined and unnecessary abbreviations that confuse the reader. I strongly recommend creating a table of abbreviations and listing them all to simplify the manuscript for a global audience.



DETAILED COMMENTS

Line 86:
You have already abbreviated "drug response prediction" as "DRP" (please see line 53). Please adhere to the use of abbreviations throughout the script.

Lines 98:
You have already abbreviated "convolution neural network" as "CNN". Please adhere to the use of abbreviations throughout the script.

Line 119:
What do you mean by GRN? Please define abbreviations clearly before citing.

Line 129:
You have already abbreviated "particle swarm optimization" as "PSO". Please adhere to the use of abbreviations throughout the script.

Line 131:
Salleh et al. (2017) They proposed → Salleh et al. (2017) proposed?

Line 132:
You are just defining GRN here, even though you have already used it.

Line 136:
What do you mean by DNN? Do you mean deep neural network?
Unfortunately, the manuscript is full of undefined and unnecessary abbreviations that confuse the reader. I strongly recommend creating a table of abbreviations and listing them all to simplify the manuscript for a global audience.

Line 143:
What do you mean by AE? Do you mean autoencoder?

Line 143:
What do you mean by RF? Do you mean random forest?

Line 144:
What do you mean by LR? Do you mean linear regression (or, maybe logistic regression)?

Line 147:
You have already abbreviated "long short-time memory" as "LSTM". Please adhere to the use of abbreviations throughout the script.

Line 152:
You are just defining DNN here, even though you have already used it.

Lines 164-165:
You have already abbreviated "particle swarm optimization" as "PSO". Please adhere to the use of abbreviations throughout the script.

Line 167:
Please convert the comma after "dataset" to a period.

Line 174:
What does the term "TGSA" stand for?

Line 177:
What does the term "MCC" stand for?

Line 187, 191-192:
You have already defined "DNN" and "CNN". Please adhere to the use of abbreviations throughout the script.

* Due to the endless number of errors in using abbreviations, I will refrain from posting such comments from now on. Please check the entire manuscript. The volume of such mistakes is very high and requires a substantial revision of the manuscript.

Line 200:
Moreover, Identifying → Moreover, identifying

Line 392:
soft max → Softmax?

Line 575:
specificity, Root mean → specificity, root mean

Lines 577 and 604:
Receiver Operating Characteristic Curve ROC →
receiver operating characteristic (ROC) curve

Reviewer 2 ·

Basic reporting

1. Writing quality:
1.1- The writing of the article is quite non-uniform. While some sentences are written clearly, many instances exist where it is not even possible to understand what the manuscript is trying to say. For example, "The ANOVA test of these two diseases can be utilized to estimate the sensitivity of selected drugs over the cancer cell lines of the gene data, like genetic feature variants, genetic features, cell line features, and drug 226 features. Table 1 shows the details of the datasets." It is unclear what the authors are trying to say. What is gene data? What is this sentence trying to convery?
1.2- As another example, the authors say "In the pre-processing stage, missing data is detached from the dataset based on a variable with more than 1/3 of the missing values." Very hard to understand what this sentence is trying to say. Also, what data modality has missing values? Gene expression in GDSC is bulk and not sparse. Also, what do you remove if there are too many missing values: cell lines, or particular feature? Also, how many features / cell lines do you start with before removing sparse ones and how many do you end up with? This needs to be clearly stated (e.g., as a supplementary table or in text).
1.3. Or they say "This removal procedure is done with the influence of the imputation process. " What does this sentence mean? Overall, there are a lot of issues with grammar, and clarity of the writing. I strongly suggest the use of a professional and scientific editing service or asking a colleague fluent in English to review and edit. Many parts of the paper needs re-writing.

2. Literature references:
2.1. The authors mention several prior work, but they are mainly quite old and more recent articles must be cited. Also, they should distinguish between methods that try to learn from cell lines to predict on cell lines (e.g., "10.1016/j.ccell.2020.09.014" ), those that learn from cell lines but make predictions on patient tumours (e.g., https://doi.org/10.1016/j.gpb.2023.01.006), and those that learn from tumours and predict on tumours. Also review papers like "https://doi.org/10.3389/fmed.2023.1086097" can help authors to find relevant citations.

3. Figures and Tables:
3.1. Figures and tables are not descriptive. For example, the quality of figures (e.g., Figure 1 , 5, 7) are too low and they are blurry.
3.2. Figures are lacking detailed caption, many are missing ylabel and xlabel (e.g., Figure 2) and overall are unclear.
3.3. Table 1 is very unclear: In table 1, it is mentioned that there are 6528 BRCA cell lines used. This implies distinct cell lines, but that is extremely unlikely. GDSC2 has only 969 cell lines in total and GSC1 has 970. What do Samples correspond to in Table 1? What do Features correspond to in Table 1? All column headers must be defined clearly. How many distinct BRCA cell lines are used? How many for PRAD?

4. definition of terms: should be significantly improved.

Experimental design

There are various issues with experimental design and even reporting of that. Many steps are unclear and are not properly explained. The data pre-processing is completely vague and even the number of samples and how the models are applied are wrong (e.g., Table 1) or misleading.

Validity of the findings

There is no benchmark and no comparison with alternative models. It is unclear how the model is implemented. Overall, I cannot recommend this article for publication.

Additional comments

1. Use of informational language (e.g. apostrophe it's) should be avoided. Also, some sentences are not fluent (e.g abstract or many other places).
2. GDSC had hundreds of CCLs (~1000). Evaluation on all of them is needed, as that is the convention and SOTA. Why only BRCA and PRAD is used? This is not ok.
3. Many references are too old (e.g., statistics on cancer from four years ago)
4. What do you mean by gene expression temporal and spatial features? Aren't these snapshot bulk?
5. gene expression at bulk is typically not sparse. Why do you need to remove sparsity?
7. How do you consolidate same drug and different names in these datasets?
8. Lines 224 to 226 don't make sense.
9. The quality of figures (e.g., Figure 1 , 5, 7) are too low.
10. In table 1, it is mentioned that there are 6528 BRCA cell lines used. This implies distinct cell lines, but that is extremely unlikely. GDSC2 has only 969 cell lines in total and GSC1 has 970. What do Samples correspond to in Table 1? What do Features correspond to in Table 1? All column headers must be defined clearly. How many distinct BRCA cell lines are used? How many for PRAD?
11. This sentences is unclear and hard to understand (needs rewording): "The ANOVA test of these two diseases can be utilized to estimate the sensitivity of selected drugs over the cancer cell lines of the gene data, like genetic feature variants, genetic features, cell line features, and drug 226 features. Table 1 shows the details of the datasets." What is "gene data"? What are "cell line features"? Overall description of features, number of them, and their nature is lacking.
12. "In the pre-processing stage, missing data is detached from the dataset based on a variable with more than 1/3 of the missing values." Very hard to understand what this sentence is trying to say. Also, what data modality has missing values? Gene expression in GDSC is bulk and not sparse. Also, what do you remove if there are too many missing values: cell lines, or particular feature? Also, how many features / cell lines do you start with before removing sparse ones and how many do you end up with? This needs to be clearly stated (e.g., as a supplementary table or in text).
13. Final Number of instances (cell lines) and features in each data modality must be clearly stated. Right now, the description is vague.
14. "This removal procedure is done with the influence of the imputation process. " What does this sentence mean? Overall, there are a lot of issues with grammar, and clarity of the writing. I strongly suggest the use of a professional and scientific editing service or asking a colleague fluent in English to review and edit. Many parts of the paper needs re-writing.
15. Figure 2 has no xlable and ylabel.
16. No benchmarking against alternative models are reported.

---

## Round 0.2 · accepted · Accept

The authors have addressed all of the reviewers' comments. Congratulations!

Reviewer 1 ·

Basic reporting

• The manuscript proposes and discusses an innovative deep learning architecture to improve cancer drug response prediction. Moreover, hyperparameters pertaining to the developed algorithm are tuned through a metaheuristic strategy to enhance the model's performance.
• The conceptualized idea is interesting and innovative and has brought state-of-the-art results for cancer-related data classification.
• The manuscript is generally well-written in terms of quality English.
• The literature is comprehensively reviewed, and updated studies in the specialized field are covered.
• The figures and tables presented are of good quality and resolution.
• The raw data used are publicly available. Moreover, the obtained results are stored in the Supplementary Materials Section.
• The obtained results are reported in a transparent and comprehensible manner.

Experimental design

• The existing challenges of traditional methods for cancer drug response prediction have been described well, and relevant ideas have been conceptualized to address them.
• The designed experiment was applied to the standard data, and the relevant results were obtained.
• Methods are described with sufficient information to be reproducible by other investigators.

Validity of the findings

• The performance of the proposed model has been evaluated through ROC analysis and diagnostic statistics thereof. The reported results indicate that the proposed model has been able to significantly improve the performance of traditional approaches in cancer drug response prediction.
• The conclusions have been appropriately stated and connected to the research questions.